# Gradient Aligned Regression via Pairwise Losses

**Dixian Zhu** [1]  **Tianbao Yang** [2]  **Livnat Jerby** [1]

## Abstract

Regression is a fundamental task in machine learning that has garnered extensive attention over the past decades. The conventional approach for regression involves employing loss functions that primarily concentrate on aligning model prediction with the ground truth for each individual data sample. Recent research endeavors have introduced novel perspectives by incorporating label similarity into regression through the imposition of additional pairwise regularization or contrastive learning on the latent feature space, demonstrating their effectiveness. However, there are two drawbacks to these approaches: (i) their pairwise operations in the latent feature space are computationally more expensive than conventional regression losses; (ii) they lack theoretical insights behind these methods. In this work, we propose GAR (Gradient Aligned Regression) as a competitive alternative method in label space, which is constituted by a conventional regression loss and two pairwise label difference losses for gradient alignment including magnitude and direction. GAR enjoys: i) the same level efficiency as conventional regression loss because the quadratic complexity for the proposed pairwise losses can be reduced to linear complexity; ii) theoretical insights from learning the pairwise label difference to learning the gradient of the ground truth function. We limit our current scope as regression on the clean data setting without noises, outliers or distributional shifts, etc. We demonstrate the effectiveness of the proposed method practically on two synthetic datasets and on eight extensive real-world tasks from six benchmark datasets with other eight competitive baselines. Running time experiments demonstrate the superior efficiency

of the proposed GAR compared to existing methods with pairwise regularization or contrastive learning in the latent feature space. Additionally, ablation studies confirm the effectiveness of each component of GAR. The code is open sourced at `https://github.com/DixianZhu/GAR`.

## 1. Introduction

As one of the most fundamental tasks in Machine Learning (ML) field, regression stands out as a powerful tool for understanding and modeling the relationships from complex data features to continuous data labels. Regression techniques have been widely utilized in many areas, such as computer vision (Moschoglou et al., 2017; Niu et al., 2016), drug discovery (Dara et al., 2022), economics and finances (Benkraiem & Zopounidis, 2021), environmental forecast (Zhu et al., 2018), material science (Stanev et al., 2018). For centuries, Mean Squared Error (MSE) and Mean Absolute Error (MAE) have been adopted to measure the distance from the ML model predictions to the ground truths. Consequently, researchers have naturally proposed to optimize the least squared error and least absolute error (Legendre, 1806; Dodge, 2008). Since the last century, to improve model stability and control model complexity, researchers have proposed various model regularizations based on MSE loss, resulting in multiple variants such as Ridge Regression, LASSO, and Elastic Net (Tikhonov et al., 1943; Tibshirani, 1996; Zou & Hastie, 2005). Along this line, the Huber loss was proposed to trade off between MAE and MSE, aiming to mitigate the issue of MSE being sensitive to outliers (Huber, 1992). All these conventional loss functions and many of their variants focus on minimizing the difference between model prediction and ground truth for each individual data sample pointwisely, but do not explicitly model and capture the relationships for multiple data samples.

Recent research has proposed incorporating label pairwise similarities into regression across different data samples, addressing part of the gap left by conventional loss functions that focus solely on individual prediction errors (Gong et al., 2022; Keramati et al., 2023; Zha et al., 2023). Those latent-space-based pairwise regression methods define an 'anchor data point', which either acts as a reference point for calculating the relative ranks to the anchor for the other

[1]Department of Genetics, Stanford University, CA, USA [2]Department of Computer Science, Texas A&M University, TX, USA. Correspondence to: Dixian Zhu <dixian-zhu@stanford.edu>, Livnat Jerby <ljerby@stanford.edu>.

*Proceedings of the 42nd International Conference on Machine Learning*, Vancouver, Canada. PMLR 267, 2025. Copyright 2025 by the author(s).

data (Gong et al., 2022), or the ranks are consequently utilized to construct positive and negative pairs (Zha et al., 2023; Keramati et al., 2023). One of the concerns for those approaches is conceptually to see: the original pairwise label similarities are converted to ranks or positive and negative pairs. There is an approximation loss because the conversion is single-directional (and the original label similarities cannot be recovered from the coarser converted information). Moreover, they also require higher computational costs than conventional loss functions, as they need to compute pairwise similarities in the latent space, which requires quadratic time related to the training batch size.

Our contributions can be summarized in three main aspects: 1) We propose pairwise losses in label space to capture pairwise label difference for regression task that can enhance rank preserving; moreover, we provide theoretical insights between learning the gradient for the ground truth function and optimizing the proposed pairwise losses. 2) We further propose equivalent and more efficient formulations for the proposed pairwise losses, which only require linear time relative to the training batch size and are as fast as conventional losses, such as MAE and MSE. 3) We demonstrate the effectiveness of the proposed method, GAR (Gradient Aligned Regression), on two synthetic datasets and eight real-world regression tasks. Extensive experiments and ablation studies are conducted for supporting that GAR is as efficient as conventional regression loss and the effectiveness of each component for GAR.

## 2. Related Work

**Regression on Generic Setting:** MSE, MAE and Huber loss are the most exposed conventional regression loss functions that work on generic regression settings (Legendre, 1806; Dodge, 2008; Huber, 1992), all of which focus on fitting the prediction to the truth for each individual data sample. Specifically, MAE and Huber loss are more robust to outliers. There are other less exposed losses for generic regression, such as Rooted Mean Squared Error (RMSE) loss and Root Mean Squared Logarithmic Error (RMSLE) loss (Ciampiconi et al., 2023), but they are also restricted to minimizing individual error. Different regularization such as $\ell_1$-norm, $\ell_2$-norm penalties on model parameters have been studied as a promising way to prevent model over-fitting and improving training stability (Tikhonov et al., 1943; Tibshirani, 1996; Zou & Hastie, 2005). Extensive research has been conducted to improve the regularization for regression by making the penalty weight adaptive or optimal under certain conditions (Zou, 2006; Wu & Xu, 2020).

**Regression on Extensive Settings:** when the data label is discrete, ordinal regression techniques can be applied to convert the regression task to either multiple binary classification tasks or one multi-class classification task (Niu et al.,

2016; Rothe et al., 2015; Zhang et al., 2023). When learning on time-series data, researchers can apply advanced ML models, such as Recurrent Neural Network (RNN), Long short-term memory (LSTM) network or Attention model, for capturing the time-series pattern (Rumelhart et al., 1986; Hochreiter & Schmidhuber, 1997; Vaswani et al., 2017); or, apply consecutive regularization that enhances the similarities for the predictions from adjacent time slots (Zhu et al., 2018). Studies propose to minimize penalties for regression on multi-task learning when there are multiple tasks or targets from the dataset (Solnon et al., 2012). Work adapts regression to active learning when there is unlabeled data available for querying and labeling (Holzmüller et al., 2023). Papers improve robustness for regression for online learning when the data is sequentially available (Pesme & Flammarion, 2020). Work improves the robustness for regression when the data is imbalanced (Yang et al., 2021; Ren et al., 2022). Research strengthens the robustness for regression when there are adversarial corruptions (Klivans et al., 2018).

**Regression with Pairwise Similarity:** recently, there are several methods are proposed to improve regression with pairwise label similarities (Gong et al., 2022; Zha et al., 2023; Keramati et al., 2023). The RankSim method is proposed to convert the label pairwise similarities for other data as ranks for each anchor data; then enforce the predictions follow the same rank by adding a MSE loss on the prediction rank versus the label rank (Gong et al., 2022). The RNC (Rank-N-Contrast) method is inspired by contrastive learning, which follows the similar fashion as SimCLR (Chen et al., 2020), but defines the positive and negative pairs by choosing different anchor data points with pairwise label similarities; RNC is proposed for pre-training that requires fine tuning afterwards (Zha et al., 2023). ConR (Contrastive Regularizer) is also proposed in contrastive learning fashion, whose positive pairs are decided by the data pairs with high label similarities and negative pairs are decided by the data pairs with high label similarities but low prediction similarities (Keramati et al., 2023). All of them impose the label pairwise similarity in latent feature space. In contrast, we propose GAR, which: 1) Directly captures the pairwise label difference in label space, theoretically connecting it to learning the gradients of the ground truth function. 2) Reduces the quadratic pairwise computation to linear complexity because the proposed pairwise losses are equivalent to the variance of prediction errors and the negative Pearson correlation coefficient between predictions and ground truths.

**Pairwise Loss in Other Applications:** pairwise loss functions have also been utilized in other fields, such as learning-to-rank, metric learning, and AUROC optimization. RankNet, for example, employs a logistic loss on sample pairs to learn the relative order of items in a ranking

function, which incurs a quadratic time complexity with respect to data size (Burges et al., 2005). In metric learning, pairwise contrastive loss (Hadsell et al., 2006) encourages embeddings of predefined similar samples to be close, while pushing dissimilar samples apart. For AUROC optimization, several pairwise loss functions have been proposed, typically computed on pairs of positive and negative samples using binary label information (Gao & Zhou, 2015). In contrast, our proposed pairwise losses are designed for regression with continuous labels, achieve linear time complexity, and do not depend on predefined similarity pairs or binary class labels for pair construction.

In this work, we focus on regression on the clean data setting without considering noises, outliers, distributional shifts, etc.

## 3. Method

Denote the underlying ground truth function for target $y \in \mathbb{R}^1$ on data $\mathbf{x} \in \mathbb{R}^d$ as $y = \mathcal{Y}(\mathbf{x})$; the training dataset as $D = \{(\mathbf{x}_1, y_1), ..., (\mathbf{x}_n, y_n)\}$, where $y_i = \mathcal{Y}(\mathbf{x}_i)$, which is sampled from underlying data distribution $\mathcal{D}$. The regression task is to learn a fitting function $f(\mathbf{x})$ that can approximate $\mathcal{Y}(\mathbf{x})$. It is straightforward to extend the single-dimensional target case to multi-dimensional target case. For the sake of simplicity, the presentation in this paper focuses on the single target form.

The conventional regression task optimizes:

$$\mathcal{L}_c = \frac{1}{N} \sum_{i=1}^{N} \ell(f(\mathbf{x}_i), y_i), \tag{1}$$

where $\ell(\cdot, \cdot)$ denote the individual loss function for each data sample. Denote the prediction error (difference between prediction and ground truth) for $\mathbf{x}$ under the model $f(\cdot)$ by: $\delta_{\mathbf{x}}^f = \delta\big(f(\mathbf{x}), \mathcal{Y}(\mathbf{x})\big) = f(\mathbf{x}) - \mathcal{Y}(\mathbf{x})$. The conventional MAE and MSE loss can be written as:

$$\mathcal{L}_c^{\text{MAE}} = \frac{1}{N} \sum_{i=1}^{N} |\delta_{\mathbf{x}_i}^f|, \quad \mathcal{L}_c^{\text{MSE}} = \frac{1}{N} \sum_{i=1}^{N} (\delta_{\mathbf{x}_i}^f)^2. \tag{2}$$

### 3.1. Motivations

Our work is motivated by the following two aspects:

1) The conventional regression loss function only focuses on minimizing the magnitude of $\delta_{\mathbf{x}}^f$, which can be blind on other evaluation perspectives. For example, there are two models $f_1(\cdot)$ and $f_2(\cdot)$ on 4 training data samples with errors as: $\{\delta_{\mathbf{x}_1}^{f_1}, ..., \delta_{\mathbf{x}_4}^{f_1}\} = \{1, -1, 1, -1\}$, $\{\delta_{\mathbf{x}_1}^{f_2}, ..., \delta_{\mathbf{x}_4}^{f_2}\} = \{1, 1, 1, 1\}$. Conventional regression loss doesn't impose any preference over $f_1(\cdot)$ or $f_2(\cdot)$. However, $f_2(\cdot)$ enjoys smaller error variance (indicating more consistent prediction and is better for rank preservation). Moreover, with

an additional bias correction, $f_2(\cdot)$ could easily achieve 0 training error.

2) Prior studies have been proposed to enforce the model to maintain the relative rank regarding to label similarities (Gong et al., 2022; Zha et al., 2023; Keramati et al., 2023). Take a simple example for age prediction, there are 3 people aged as $y_1 = 10$, $y_2 = 20$, $y_3 = 70$; the prior methods either impose a regularization or adopt contrastive learning in the latent feature space with pairwise fashion to push prediction $f(\mathbf{x}_2)$ to be closer to $f(\mathbf{x}_1)$ than $f(\mathbf{x}_3)$.

Motivated by the second point, we explicitly enforce the pairwise prediction difference $f(\mathbf{x}_i) - f(\mathbf{x}_j)$ to be close to the pairwise label difference $y_i - y_j$ through two pairwise losses in the label space: the first directly matches the original label difference, while the second is a relaxed version with a scaling factor. Next, we prove that: the first pairwise loss is equivalent to the variance of prediction errors and the second is equivalent to negative Pearson correlation coefficient between predictions and ground truths, which addresses the concern in the first point. Moreover, the equivalent forms reduce the quadratic time complexity for the proposed pairwise losses to linear complexity. Last but not least, we provide theoretical insight between learning the pairwise label difference and learning the gradients of the ground truth function.

### 3.2. Pairwise Losses

We propose the following pairwise loss of label difference for a regression task:

$$\mathcal{L}_{\text{diff}} = \frac{1}{N^2} \sum_{i=1}^{N} \sum_{j=1}^{N} \ell(f(\mathbf{x}_i) - f(\mathbf{x}_j), \mathcal{Y}(\mathbf{x}_i) - \mathcal{Y}(\mathbf{x}_j)), \tag{3}$$

where the individual loss $\ell(\cdot, \cdot)$ is defined similarly as conventional regression. $\mathcal{L}_{\text{diff}}$ explicitly and directly enforces the pairwise prediction difference to be close to the pairwise label difference.

**Theorem 1** *When choose individual loss $\ell$ as $\frac{1}{2}$ squared error function, $\ell_{\frac{1}{2}MSE}(a, b) = \frac{1}{2}(a - b)^2$:*

$$\mathcal{L}_{diff}^{MSE} := \sum_{i=1}^{N} \sum_{j=1}^{N} \frac{\ell_{\frac{1}{2}MSE}(f(\mathbf{x}_i) - f(\mathbf{x}_j), \mathcal{Y}(\mathbf{x}_i) - \mathcal{Y}(\mathbf{x}_j))}{N^2}$$
$$= Var(\delta_{\mathbf{x}}^f), \tag{4}$$

*where $Var(\cdot)$ denote the empirical variance. The proof is included in Appendix A.1.*

The proof is based on simple algebras that add and subtract the mean value of $\delta_{\mathbf{x}}^f$, denoted as $\bar{\delta}_{\mathbf{x}}^f$. The similar proof logic

can be used to decouple MSE loss, $\mathcal{L}_c^{\text{MSE}} = \frac{1}{N} \sum_{i=1}^{N} (\delta_{\mathbf{x}_i}^f)^2$:

$$\mathcal{L}_c^{\text{MSE}} = \frac{1}{N} \sum_{i=1}^{N} (\delta_{\mathbf{x}_i}^f - \bar{\delta}_{\mathbf{x}}^f + \bar{\delta}_{\mathbf{x}}^f)^2$$

$$= \frac{1}{N} \sum_{i=1}^{N} [(\delta_{\mathbf{x}_i}^f - \bar{\delta}_{\mathbf{x}}^f)^2 + (\bar{\delta}_{\mathbf{x}}^f)^2] = \text{Var}(\delta_{\mathbf{x}}^f) + (\bar{\delta}_{\mathbf{x}}^f)^2.$$

MSE lacks the flexibility to trade off the mean and variance of errors. Simply summing them results in the mean of squared individual errors, which loses the scope of group patterns.

**Corollary 2** *When $\ell$ is $\frac{1}{2}$ squared error, the loss of pairwise label difference has the following simpler and more efficient empirical form:*

$$\mathcal{L}_{diff}^{MSE} = \frac{1}{N} \sum_{i=1}^{N} \left( (f(\mathbf{x}_i) - \bar{f}) - (y_i - \bar{y}) \right)^2. \quad (5)$$

*The corollary directly comes from Theorem 1 by the definition of variance.*

It is obvious that $\mathcal{L}_{\text{diff}}^{\text{MSE}}$ enjoys a linear time complexity. Next, we propose a relaxed version for $\mathcal{L}_{\text{diff}}$. We apply a relaxing scaling factor when match the pairwise differences from predictions and ground truths, in order to allow mismatch on magnitude:

$$f(\mathbf{x}_i) - f(\mathbf{x}_j) = C \cdot [\mathcal{Y}(\mathbf{x}_i) - \mathcal{Y}(\mathbf{x}_j)].$$

Denote the pairwise difference as $df_{i,j} = f(\mathbf{x}_i) - f(\mathbf{x}_j)$ and $d\mathcal{Y}_{i,j} = \mathcal{Y}(\mathbf{x}_i) - \mathcal{Y}(\mathbf{x}_j)$ for the prediction and the ground truth, the vector notations for them as $\mathbf{df} = [df_{1,1}, df_{1,2}, ..., df_{N,N-1}, df_{N,N}]$, $\mathbf{d\mathcal{Y}} = [d\mathcal{Y}_{1,1}, d\mathcal{Y}_{1,2}, ..., d\mathcal{Y}_{N,N-1}, d\mathcal{Y}_{N,N}]$. We proposed the following p-norm loss function for the **normalized** pairwise difference to capture ground truth function shape, where the scaling factor is removed by normalization:

$$\mathcal{L}_{\text{diffnorm}} = \| \frac{\mathbf{df}}{\|\mathbf{df}\|_p} - \frac{\mathbf{d\mathcal{Y}}}{\|\mathbf{d\mathcal{Y}}\|_p} \|_p^p. \quad (6)$$

**Corollary 3** *When choose $p = 2$ for $\mathcal{L}_{diffnorm}$, it equals to: $1-$ Pearson correlation coefficient:*

$$\mathcal{L}_{diffnorm}^{p=2} = \frac{1}{2N} \sum_{i=1}^{N} \left( \frac{(f(\mathbf{x}_i) - \bar{f})}{\sqrt{Var(f)}} - \frac{(y_i - \bar{y})}{\sqrt{Var(y)}} \right)^2$$

$$= 1 - \frac{Cov(f, y)}{\sqrt{Var(f)Var(y)}}$$

$$= 1 - \rho(f, y). \quad (7)$$

*where $Cov(\cdot, \cdot)$ denotes covariance and $\rho(\cdot, \cdot)$ denotes Pearson correlation coefficient. We include the proof in the Appendix A.2. The formulation also enjoys linear time complexity.*

Next, we provide the theoretical insight between learning pairwise label difference and learning the gradients of ground truth function.

**Theorem 4** *For two K-order differentiable deterministic functions with open domain $f(\cdot) : \mathbb{R}^d \to \mathbb{R}^1$, $\mathcal{Y}(\cdot) : \mathbb{R}^d \to \mathbb{R}^1$, the following holds: $f(\mathbf{x}_1) - f(\mathbf{x}_2) = \mathcal{Y}(\mathbf{x}_1) - \mathcal{Y}(\mathbf{x}_2)$, $\forall (\mathbf{x}_1, \mathcal{Y}(\mathbf{x}_1)), (\mathbf{x}_2, \mathcal{Y}(\mathbf{x}_2)) \in \mathcal{D}$, **iff** $\nabla^k f(\mathbf{x}) = \nabla^k \mathcal{Y}(\mathbf{x})$, $\forall (\mathbf{x}, \mathcal{Y}(\mathbf{x})) \in \mathcal{D}$, $k = \{1, 2, ..., K\}$.*

*We include proof in the Appendix A.3.*

**Remark 1:** The proof mainly depends on Mean Value Theorem, Limits and L'Hôpital's rule. Besides of fitting the prediction $f(\mathbf{x}_i)$ to be close to the ground truth $y_i$, that is, $f(\mathbf{x}) \approx \mathcal{Y}(\mathbf{x})$, $\forall (\mathbf{x}, \mathcal{Y}(\mathbf{x})) \in \mathcal{D}$, we additionally make the model to explicitly capture the gradient (and the higher-order gradients) of the ground truth function, i.e. $\nabla^k f(\mathbf{x}) \approx \nabla^k \mathcal{Y}(\mathbf{x})$, $k = \{1, 2, ...\}$, with the proposed pairwise losses.

**Remark 2:** Although we assume that the function is deterministic for simplicity, this theorem is also applicable to common stochastic functions in a certain form. For example, $\mathcal{Y}(\mathbf{x}, z) = m(\mathbf{x}) + z \cdot s(\mathbf{x})$, where $z \sim N(0, 1)$ is stochastically sampled from the standard normal distribution (also works for any zero-mean distribution). Let $m(\mathbf{x})$ and $s(\mathbf{x})$ represent the ground truth mean and standard deviation for the general heteroskedastic case. We can still prove that the model can capture the expected gradient in $\mathbf{x}$, that is, $\mathbb{E}_z [\nabla_{\mathbf{x}} \mathcal{Y}(\mathbf{x}, z)]$, if and only if the difference in the prediction pairs matches the difference in the truth pairs under expectation. We have the following extended corollary.

**Corollary 5** *For K-order differentiable deterministic function with open domain $f(\cdot) : \mathbb{R}^d \to \mathbb{R}^1$, and K-order differentiable stochastic function with open domain $\mathcal{Y}(\cdot, \cdot) : \mathbb{R}^{d+1} \to \mathbb{R}^1$ that can be written as $\mathcal{Y}(\mathbf{x}, z) = m(\mathbf{x}) + z \cdot s(\mathbf{x})$, where $z$ is randomly sampled from a distribution with zero-mean and $m(\cdot) : \mathbb{R}^d \to \mathbb{R}^1$, $s(\cdot) : \mathbb{R}^d \to \mathbb{R}^1$. The following holds: $f(\mathbf{x}_1) - f(\mathbf{x}_2) = \mathbb{E}_{z_1} [\mathcal{Y}(\mathbf{x}_1, z_1)] - \mathbb{E}_{z_2} [\mathcal{Y}(\mathbf{x}_2, z_2)]$, $\forall z_1, z_2 \sim N(0, 1)$; $\forall (\mathbf{x}_1, \mathcal{Y}(\mathbf{x}_1, z_1)), (\mathbf{x}_2, \mathcal{Y}(\mathbf{x}_2, z_2)) \in \mathcal{D}$, **iff** $\nabla^k f(\mathbf{x}) = \mathbb{E}_z [\nabla_{\mathbf{x}}^k \mathcal{Y}(\mathbf{x}, z)]$, $\forall (\mathbf{x}, \mathcal{Y}(\mathbf{x}, z)) \in \mathcal{D}$; $\forall z \sim N(0, 1)$.*

**Proof sketch:** $\mathbb{E}_z [\mathcal{Y}(\mathbf{x}, z)] = m(\mathbf{x})$, $\mathbb{E}_z [\nabla_{\mathbf{x}}^k \mathcal{Y}(\mathbf{x}, z)] = \nabla^k m(\mathbf{x})$. Our previous proof for Theorem 4 still goes through by replacing $\mathcal{Y}(\mathbf{x})$ with $m(\mathbf{x})$. Notice that the stochastic term $z \cdot s(\mathbf{x})$ can be canceled by taking the expectation over $z$ due to zero-mean.

**Remark 3:** It should be noted that the proposed pairwise losses are designed to work with conventional pointwise regression loss (e.g. MAE), which complementarily captures pointwise function values.

### 3.3. Efficient and Robust Aggregation for Different Losses

Now that we have three losses that are important for the proposed GAR (Gradient Aligned Regression): $\mathcal{L}_c$, $\mathcal{L}_{\text{diff}}$, and $\mathcal{L}_{\text{diffnorm}}$; more specifically, we will focus on $\mathcal{L}_c^{\text{MAE}}, \mathcal{L}_{\text{diff}}^{\text{MSE}}, \mathcal{L}_{\text{diffnorm}}^{p=2}$ in the rest of the presentation. Next, we propose a robust approach to aggregate them.

Given the huge diversity of the dataset and model, it is time-consuming to tune the three terms with a convex combination. Furthermore, the magnitudes for each term can be different, which adds more difficulty to balance them. For example, $\mathcal{L}_{\text{diffnorm}}^{p=2} \in [0, 2]$ but $\mathcal{L}_c^{\text{MAE}}$ can be infinitely large; therefore, $\mathcal{L}_{\text{diffnorm}}^{p=2}$ could be easily overwhelmed if simply combining other components with larger magnitudes by arithmetic mean, i.e. $\mathcal{L} = (\mathcal{L}_c^{\text{MAE}} + \mathcal{L}_{\text{diff}}^{\text{MSE}} + \mathcal{L}_{\text{diffnorm}}^{p=2})/3$.

An intuitive way to amplify the significance of $\mathcal{L}_{\text{diffnorm}}^{p=2}$ is the geometric mean, that is, $\mathcal{L} = (\mathcal{L}_c^{\text{MAE}} \mathcal{L}_{\text{diff}}^{\text{MSE}} \mathcal{L}_{\text{diffnorm}}^{p=2})^{1/3}$, where the effects of different loss magnitudes are unified. However, it could raise another issue. Consider $\mathcal{L}(a, b, c) = (abc)^{1/3}$, $\frac{\partial \mathcal{L}(a,b,c)}{\partial a} = \frac{(bc)^{1/3}}{a^{2/3}}$. The smaller component can get a larger gradient but may experience a numerical issue when $a \ll bc$. As a consequence, the overall objective could focus solely on the smallest component.

Inspired by the previous two intuitive examples (the loss with smaller magnitude either gains too less attention or too much attention), we propose an efficient and robust aggregation approach for GAR based on a variant of Distributionally Robust Optimization (DRO), which can not only trade off the previous two cases, but also only requires one tuning hyperparameter (Zhu et al., 2023). For each loss, we apply a logarithmic transformation to reduce the magnitude effect. For the sake of generality, we denote the loss as $\mathcal{L}_i$ and the overall loss as $\mathcal{L}_{\text{GAR}}(\mathcal{L}_1, ..., \mathcal{L}_M)$, where the overall loss is composed with $M$ sub-losses. Denote $D(\mathbf{p}|\frac{1}{M})$ as divergence measure from probability vector $\mathbf{p}$ over simplex $\Delta_M$ to the uniform distribution $\frac{1}{M}$. The DRO formulation for GAR takes the balance from the averaged value to the maximal value:

$$\mathcal{L}_{\text{GAR}}(\mathcal{L}_1, ..., \mathcal{L}_M; \alpha) = \max_{\mathbf{p} \in \Delta_M} \sum_{i=1}^{M} p_i \log \mathcal{L}_i - \alpha D(\mathbf{p}|\frac{1}{M}), \tag{8}$$

where $\alpha \geq 0$ is the robust hyper-parameter for GAR.

**Theorem 6** *When take $D(\cdot|\cdot)$ as KL-divergence, GAR has the following equivalent formulation:*

$$\mathcal{L}_{GAR}^{KL}(\mathcal{L}_1, ..., \mathcal{L}_M; \alpha) = \alpha \log(\frac{1}{M} \sum_{i=1}^{M} \mathcal{L}_i^{1/\alpha}). \tag{9}$$

*It is worth noting that:*

$$\exp\left(\mathcal{L}_{GAR}^{KL}(\mathcal{L}_1, ..., \mathcal{L}_M; \alpha)\right) = (\frac{1}{M} \sum_{i=1}^{M} \mathcal{L}_i^{1/\alpha})^{\alpha}$$

$$= \begin{cases} (\Pi_{i=1}^{M} \mathcal{L}_i)^{\frac{1}{M}}, & \alpha \to +\infty. \\ \frac{1}{M} \sum_{i=1}^{M} \mathcal{L}_i, & \alpha = 1. \\ \arg\max_{i=1}^{M} \mathcal{L}_i, & \alpha \to 0. \end{cases} \tag{10}$$

*which trade off from geometric mean, arithmetic mean and the maximal value for $\mathcal{L}_1, ..., \mathcal{L}_M$.*

**Remark:** the proof is included in the Appendix A.4.

Next, we discuss how to take care of extreme case for $\mathcal{L}_{\text{GAR}}$. Notice that $\mathcal{L}_i^{1/\alpha}$ can have numerical issue when $\alpha \to 0, \mathcal{L}_i \to +\infty$ or $\alpha \to +\infty, \mathcal{L}_i \to 0$. The former case can cause computation overflow on forward propagation and the later case can cause computation overflow on backward propagation. Denote $\mathcal{L}_{\max} = \arg\max_{i=1}^{M} \mathcal{L}_i$ and $\mathcal{L}_{\min} = \arg\min_{i=1}^{M} \mathcal{L}_i$. We utilize the constant value (detached from back propagation) of $\mathcal{L}_{\max}$ or $\mathcal{L}_{\min}$ to control $\mathcal{L}_{\text{GAR}}^{\text{KL}}$. It is easy to show $\mathcal{L}_{\text{GAR}}^{\text{KL}}$ has the following equivalent empirical formulation:

$$\mathcal{L}_{\text{GAR}}^{\text{KL}} = \begin{cases} \alpha \log\left(\frac{1}{M} \sum_{i=1}^{M} (\frac{\mathcal{L}_i}{\mathcal{L}_{\max}})^{1/\alpha}\right) + \log \mathcal{L}_{\max}, & \alpha < 1. \\ \alpha \log\left(\frac{1}{M} \sum_{i=1}^{M} (\frac{\mathcal{L}_i}{\mathcal{L}_{\min}})^{1/\alpha}\right) + \log \mathcal{L}_{\min}, & \alpha \geq 1. \end{cases} \tag{11}$$

by which we can avoid the numerical issue for both forward and backward propagation. We present algorithm pseudocode in Alg.1.

---

**Algorithm 1** Gradient Aligned Regression (GAR)

---

**Require:** hyper-parameter $\alpha$ for balancing sub-losses, training dataset $D = \{(\mathbf{x}_i, y_i)_{i=1}^{N}\}$.
  Initialize model $f(\cdot)$.
  **for** t = 1 to T **do**
    Sample mini-batch of data $\{(\mathbf{x}_i, y_i)_{i \in B_t}\}$.
    Compute MAE loss $\mathcal{L}_c^{\text{MAE}}$.
    Compute the losses for derivative: $\mathcal{L}_{\text{diff}}^{\text{MSE}}$ and $\mathcal{L}_{\text{diffnorm}}^{p=2}$ by E.q. 5 and E.q. 7.
    Compute GAR (KL) loss $\mathcal{L}_{\text{GAR}}^{\text{KL}}$ by E.q. 11.
    Utilize SGD or Adam optimizers to optimize model with gradient $\nabla_f \mathcal{L}_{\text{GAR}}^{\text{KL}}$.
  **end for**

---

**Remark:** the computational complexity for Alg.1 is $O(TB)$, where $T$ is iteration number and $B$ is batch size.

## 4. Experiments

We demonstrate the effectiveness of GAR with two synthetic datasets, eight real-world tasks on five tabular benchmark

datasets and one image benchmark dataset. Further analysis for running time of GAR, ablation studies on different components, sensitivity to hyper-parameter $\alpha$ and batch sizes are presented afterwards.

## 4.1. Synthetic Experiments

We first demonstrate the effectiveness of GAR on two synthetic datasets as toy examples.

**Datasets:** 1) Sine: we take $x_i$ from [-10$\pi$, 10$\pi$] with the interval 0.1 and $y_i = \sin(x_i)$. The data ground truth is presented as the grey solid line in Fig 1. 2) Squared sine: we take $\tilde{x}_i$ from $-1024$ to $1024$ with the interval 0.1, then calculate $x_i = \text{sign}(\tilde{x}_i)\sqrt{|\tilde{x}_i|}$, which increases the density of data from the region far from the origin. Finally, denote $\overline{x_i^2}$ as the empirical mean value of $x_i^2$, make $y_i = x_i^2 \sin(x_i)/\overline{x_i^2}$, which magnifies the target values that are far from the origin. The data ground truth is presented as the solid grey line in Fig 2.

**Baselines:** we compare the proposed GAR with 4 approaches: MAE, MSE, RNC and an extra heuristic fused loss (named MAE-Pearson) that combines MAE with negative Pearson correlation coefficient linearly:

$$\mathcal{L}_{\text{MAE}-\rho} = \beta\mathcal{L}_c^{\text{MAE}} + (1-\beta)\mathcal{L}_{\text{diffnorm}}^{p=2},$$

where recall that $\mathcal{L}_{\text{diffnorm}}^{p=2} = 1 - \rho(f, y)$, and $\beta$ is dynamically assigned as the constant value of $\rho(f, y)$ on each iteration. Intuitively, when the Pearson correlation coefficient is high, the model focuses more on the MAE loss; and vice versa. It is worth noting that GAR can also be adapted to Symbolic Regression (SR) settings. For example, GAR loss could be defined as a fitness function for evolutionary algorithm based method or reward function for reinforcement learning based method (Cranmer, 2023; Petersen et al., 2019). However, given that this work focuses on Numeric Regression (NR) setting, it is unfair to directly compare the proposed GAR under the NR setting with SR methods here. Because SR methods have placed the ground truth functions in their 'toolbox' (as search candidates) for these toy examples. To elaborate on this point, we extensively employ a well-established SR method, PySR[1] on the synthetic datasets. We find PySR performs perfectly when trigonometric functions are included in the search space. However, it performs much worse when we exclude trigonometric functions. Results are presented in the Appendix B.4.

**Model:** We utilize a 7-layer simple Feed Forward Neural Network (FFNN) with 5 hidden layers where neuron number for each layer is set as 100 with ELU activation function (Clevert et al., 2015).

**Experimental Settings:** For both of the synthetic datasets,

we uniformly randomly sample half of the data as the training dataset. The total training epochs are set as 300 and batch size is set as 128. The initial learning rate is tuned in {1e-1, 1e-2, 1e-3, 1e-4} for Adam optimizer (Kingma & Ba, 2014), which is stage-wised decreasing by 10 folds at the end of the 100-th and 200-th epoch. The weight decay is tuned in {1e-3,1e-4,1e-5,0}. RNC takes the first 100 epochs are pre-training and the remaining as fine-tuning with MAE loss; temperature for RNC is tuned in {1,2,4}. For each baseline, we run 5 trials independently with the following 5 seeds: {1,2,3,4,5}. We set the $\alpha = 0.5$ for GAR on all synthetic experiments.

**Results:** the results on the sine dataset are summarized in Fig. 1 and the results on the squared sine dataset are summarized in Fig. 2. The figures for mean and standard deviation of the predictions are shown on the right. Due to the limited space, we include MSE results in Appendix B.3 as it is similar with MAE. On both of the datasets, we can see there is a clear advantage for the proposed GAR over the conventional MSE/MAE loss. For the sine dataset, GAR captures approximately 1 or 2 more peaks than MSE/MAE loss. For the squared sine dataset, GAR can almost recover all the patterns (both shape and magnitude) of the ground truth; however, MSE/MAE loss is hard to capture the pattern of the ground truth except a part of peaks with the largest magnitude. Compared with $\mathcal{L}_{\text{MAE}-\rho}$ and RNC, GAR also demonstrates clear advantages on the both synthetic datasets for capturing more peaks on the sine dataset and capturing magnitudes better on the squared sine dataset. We also conduct experiments on the sine dataset with different number of layers for FFNN model in Appendix B.5, Fig 6, which again demonstrates similar results.

## 4.2. Benchmark Experiments

We conduct experiments for comparing GAR with other eight competitive baselines on eight real-world tasks.

**Datasets and Tasks:** 1) Concrete Compressive Strength (Yeh, 1998): predicting the compressive strength of high-performance concrete. 2) Wine Quality (Cortez et al., 2009): predicting wine quality based on physicochemical test values (such as acidity, sugar, chlorides, etc). 3) Parkinson (Total) (Tsanas et al., 2009): predicting clinician's total unified Parkinson's disease rating scale (UPDRS) score by biomedical voice measurements from 42 people with early-stage Parkinson's disease recruited to a six-month trial of a telemonitoring device for remote symptom progression monitoring. 4) Parkinson (Motor) (Tsanas et al., 2009): predicting clinician's motor UPDRS score with the same previous data feature. 5) Super Conductivity (Hamidieh, 2018): predicting the critical temperature for super conductors with 81 extracted material features. 6) IC50 (Garnett et al., 2012): predicting drugs' half-maximal inhibitory con-

---

[1] https://astroautomata.com/PySR/

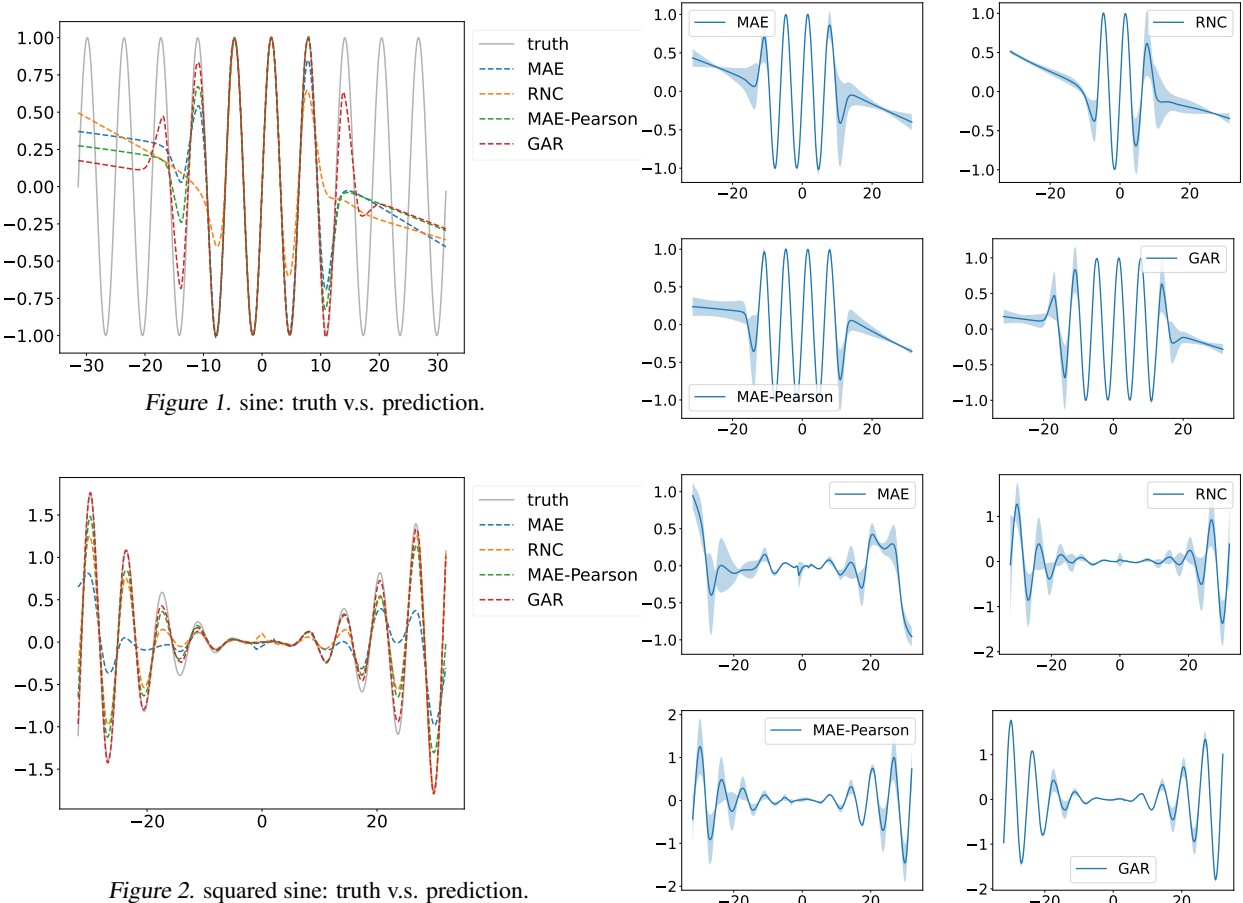

Figure 1. sine: truth v.s. prediction.

Figure 2. squared sine: truth v.s. prediction.

centration (IC50) for 15 drugs filtered from 130 total drugs with missing-value-control over 639 human tumour cell lines. We include more details about data preprocessing in the appendix. 7) AgeDB (Scratch) (Moschoglou et al., 2017): predicting age from face images. All baselines are trained from scratch. 8) AgeDB (RNC Linear Probe): predicting age with the latent features from a pre-trained RNC model (Zha et al., 2023). The statistics of the datasets are summarized in Table 3 in Appendix B.1.

**Baselines:** the baselines for comparison are: 1) MAE. 2) MSE. 3) Huber loss (Huber, 1992). 4) Focal (MAE) (Yang et al., 2021): an variant for Focal loss with MAE loss, which put more learning weights on harder samples. 5) Focal (MSE) (Yang et al., 2021): combine Focal loss with MSE loss similarly as Focal (MAE). 6) RankSim (Gong et al., 2022): a regression method that regularizes predictions to maintain similar order with the their labels. 7) RNC (Zha et al., 2023): a state-of-the-art (SOTA) contrastive learning based pre-training method for regression that enhances the prior that the pairwise similarities from learned latent features should be close to pairwise similarities from data labels. 8) ConR (Keramati et al., 2023): a contrastive learning

based SOTA regression method that combines conventional regression loss with a contrastive loss that pushes data latent features to be closer if their labels are closer. 9) GAR: the proposed Gradient Aligned regression method via pairwise losses.

**Models:** for the tabular datasets with feature dimension less equal to 16, we utilize a 6-layer-FFNN with hidden neurons as (16, 32, 16, 8) as the backbone model. For the tabular datasets with feature dimension larger to 16 and less equal to 128, we utilize a 6-layer-FFNN with hidden neurons as (128, 256, 128, 64) as the backbone model; for IC50, we generate 15 linear heads for predicting the 15 targets as the last prediction layer. All FFNN variants utilize ELU as the activation in this work. For AgeDB (Scratch), we follow the previous work and employ ResNet18 as the backbone model (Zha et al., 2023). For AgeDB (RNC Linear Probe), we simply train a linear model based on the latent feature pre-trained with RNC.

**Experimental Settings:** we follow the standard procedure for data splitting, cross validation and hyper-parameter tuning. The full details are included in Appendix B.6.

*Table 1.* Testing performance on eight different datasets/settings. Mean and standard deviation are reported. 'Gains' includes the percentages that GAR outperform the MAE and the best competitor (in parenthesis). The 'p-values' are calculated based on Student's t-test for the 'Gains'. p-value < 0.05 usually means significant difference. ↓ means the smaller the better; ↑ means the larger the better.

| Dataset | Method | MAE | MSE | Huber | Focal (MAE) | Focal (MSE) | RankSim | RNC | ConR | GAR (ours) | Gains (%) | p-values |
|---|---|---|---|---|---|---|---|---|---|---|---|---|
| Concrete Compressive Strength | MAE ↓ | 4.976(0.071) | 4.953(0.139) | 4.698(0.248) | 4.971(0.069) | 4.98(0.122) | 11.753(0.604) | 4.776(0.106) | 5.016(0.131) | **4.603(0.075)** | 7.48(2.02) | 0.0(0.48) |
| | RMSE ↓ | 6.639(0.092) | 6.549(0.221) | 6.329(0.226) | 6.622(0.071) | 6.549(0.237) | 14.503(0.716) | 6.383(0.076) | 6.693(0.155) | **6.222(0.148)** | 6.28(1.69) | 0.0(0.45) |
| | Pearson ↑ | 0.919(0.002) | 0.918(0.008) | 0.922(0.005) | 0.919(0.002) | 0.919(0.006) | 0.673(0.06) | 0.923(0.002) | 0.917(0.004) | **0.929(0.004)** | 1.14(0.61) | 0.0(0.03) |
| | Spearman ↑ | 0.917(0.004) | 0.923(0.003) | 0.925(0.004) | 0.918(0.004) | 0.921(0.006) | 0.703(0.039) | 0.926(0.003) | 0.918(0.004) | **0.931(0.003)** | 1.46(0.48) | 0.0(0.09) |
| Wine Quality | MAE ↓ | 0.5(0.005) | 0.55(0.004) | 0.536(0.004) | 0.518(0.004) | 0.552(0.003) | 0.52(0.004) | 0.545(0.004) | 0.504(0.005) | **0.494(0.004)** | 1.09(1.09) | 0.1(0.1) |
| | RMSE ↓ | 0.715(0.006) | 0.707(0.005) | 0.693(0.005) | 0.696(0.007) | 0.705(0.003) | 0.715(0.009) | 0.71(0.004) | 0.712(0.01) | **0.69(0.004)** | 3.53(0.46) | 0.0(0.32) |
| | Pearson ↑ | 0.573(0.01) | 0.585(0.006) | 0.606(0.008) | 0.602(0.009) | 0.589(0.005) | 0.581(0.011) | 0.584(0.006) | 0.586(0.012) | **0.613(0.011)** | 7.02(1.05) | 0.0(0.36) |
| | Spearman ↑ | 0.602(0.01) | 0.607(0.006) | 0.622(0.011) | 0.622(0.006) | 0.606(0.005) | 0.607(0.01) | 0.607(0.005) | 0.614(0.007) | **0.635(0.007)** | 5.58(2.13) | 0.0(0.07) |
| Parkinson (Total) | MAE ↓ | 2.875(0.862) | 3.464(0.029) | **2.442(0.379)** | 2.693(0.91) | 3.481(0.042) | 8.616(0.007) | 2.948(0.079) | 3.581(0.519) | 2.64(0.071) | 8.15(-8.14) | 0.6(0.33) |
| | RMSE ↓ | 4.803(0.809) | 5.175(0.149) | 4.498(0.392) | 4.744(0.82) | 5.157(0.14) | 10.787(0.022) | 4.786(0.103) | 5.431(0.359) | **4.258(0.206)** | 11.36(5.33) | 0.23(0.31) |
| | Pearson ↑ | 0.891(0.038) | 0.876(0.008) | 0.909(0.017) | 0.894(0.038) | 0.877(0.007) | 0.125(0.042) | 0.896(0.004) | 0.862(0.019) | **0.922(0.009)** | 3.44(1.44) | 0.15(0.21) |
| | Spearman ↑ | 0.87(0.047) | 0.856(0.008) | 0.895(0.02) | 0.874(0.047) | 0.855(0.006) | 0.141(0.051) | 0.876(0.002) | 0.842(0.025) | **0.907(0.01)** | 4.24(1.36) | 0.16(0.31) |
| Parkinson (Motor) | MAE ↓ | 1.846(0.494) | 2.867(0.092) | 1.903(0.282) | **1.599(0.047)** | 2.86(0.081) | 6.891(0.013) | 2.299(0.068) | 2.698(0.466) | 1.652(0.069) | 10.52(-3.3) | 0.46(0.24) |
| | RMSE ↓ | 3.336(0.491) | 4.06(0.067) | 3.226(0.35) | 2.992(0.105) | 4.083(0.116) | 8.082(0.019) | 3.632(0.136) | 4.193(0.331) | **2.68(0.134)** | 19.68(10.45) | 0.03(0.01) |
| | Pearson ↑ | 0.91(0.029) | 0.865(0.005) | 0.917(0.019) | 0.93(0.005) | 0.863(0.008) | 0.125(0.043) | 0.894(0.008) | 0.854(0.023) | **0.943(0.006)** | 3.71(1.48) | 0.05(0.01) |
| | Spearman ↑ | 0.902(0.032) | 0.854(0.004) | 0.912(0.019) | 0.924(0.006) | 0.854(0.009) | 0.144(0.036) | 0.885(0.008) | 0.843(0.024) | **0.939(0.006)** | 4.16(1.63) | 0.05(0.01) |
| Super Conductivity | MAE ↓ | 8.365(0.027) | 7.717(0.014) | 7.42(0.052) | 8.351(0.025) | 7.713(0.008) | 11.99(0.403) | **6.238(0.078)** | 8.368(0.022) | 6.257(0.027) | 25.2(-0.31) | 0.0(0.65) |
| | RMSE ↓ | 13.677(0.061) | 12.158(0.113) | 12.639(0.067) | 13.631(0.039) | 12.13(0.107) | 17.089(0.408) | 11.106(0.18) | 13.68(0.084) | **10.745(0.183)** | 21.44(3.25) | 0.0(0.02) |
| | Pearson ↑ | 0.918(0.0) | 0.935(0.001) | 0.93(0.001) | 0.918(0.0) | 0.935(0.0) | 0.867(0.007) | 0.947(0.001) | 0.918(0.001) | **0.949(0.002)** | 3.39(0.24) | 0.0(0.06) |
| | Spearman ↑ | 0.915(0.001) | 0.924(0.0) | 0.93(0.001) | 0.916(0.001) | 0.925(0.0) | 0.86(0.008) | 0.943(0.001) | 0.917(0.001) | **0.944(0.002)** | 3.13(0.15) | 0.0(0.18) |
| IC50 | MAE ↓ | 1.364(0.002) | 1.382(0.004) | 1.366(0.004) | 1.365(0.001) | 1.386(0.01) | 1.384(0.009) | 2.143(0.242) | **1.34(0.025)** | 1.359(0.003) | 0.37(-1.43) | 0.02(0.16) |
| | RMSE ↓ | 1.706(0.013) | 1.699(0.008) | 1.702(0.002) | 1.696(0.003) | 1.707(0.014) | 1.809(0.13) | 2.454(0.235) | **1.668(0.022)** | 1.7(0.005) | 0.39(-1.89) | 0.38(0.02) |
| | Pearson ↑ | 0.085(0.049) | 0.021(0.013) | 0.069(0.061) | 0.104(0.029) | 0.023(0.017) | 0.012(0.026) | 0.018(0.05) | 0.202(0.033) | **0.302(0.028)** | 252.97(49.13) | 0.0(0.0) |
| | Spearman ↑ | 0.114(0.032) | 0.065(0.039) | 0.087(0.054) | 0.109(0.033) | 0.064(0.04) | 0.033(0.028) | 0.038(0.05) | 0.184(0.04) | **0.26(0.024)** | 127.64(40.86) | 0.0(0.01) |
| AgeDB (Scratch) | MAE ↓ | 6.449(0.058) | 6.517(0.097) | 6.387(0.046) | 6.404(0.037) | 6.567(0.107) | 6.502(0.085) | — | 6.43(0.028) | **6.286(0.034)** | 2.52(1.58) | 0.0(0.01) |
| | RMSE ↓ | 8.509(0.078) | 8.608(0.046) | 8.418(0.045) | 8.476(0.067) | 8.617(0.104) | 8.596(0.188) | — | 8.49(0.049) | **8.268(0.05)** | 2.83(1.79) | 0.0(0.0) |
| | Pearson ↑ | 0.917(0.002) | 0.914(0.001) | 0.918(0.001) | 0.918(0.002) | 0.914(0.001) | 0.916(0.005) | — | 0.916(0.002) | **0.921(0.001)** | 0.48(0.35) | 0.0(0.0) |
| | Spearman ↑ | 0.919(0.002) | 0.917(0.001) | 0.92(0.0) | 0.92(0.002) | 0.916(0.001) | 0.918(0.004) | — | 0.919(0.003) | **0.923(0.002)** | 0.41(0.28) | 0.01(0.01) |
| | R² ↑ | 0.836(0.003) | 0.833(0.002) | 0.84(0.002) | 0.838(0.003) | 0.832(0.004) | 0.833(0.007) | — | 0.837(0.002) | **0.846(0.002)** | 1.15(0.67) | 0.0(0.0) |
| AgeDB (RNC Linear Probe) | MAE ↓ | 6.124(0.039) | 6.13(0.058) | 6.124(0.086) | 6.102(0.039) | 6.139(0.033) | — | — | — | **6.069(0.022)** | 0.89(0.55) | 0.04(0.17) |
| | RMSE ↓ | 8.077(0.036) | 8.099(0.04) | 8.108(0.068) | 8.073(0.029) | 8.113(0.037) | — | — | — | **8.054(0.014)** | 0.28(0.23) | 0.28(0.29) |
| | Pearson ↑ | 0.924(0.0) | 0.924(0.0) | 0.924(0.0) | 0.924(0.0) | 0.924(0.0) | — | — | — | 0.924(0.0) | 0.0(0.0) | 1.0(1.0) |
| | Spearman ↑ | 0.927(0.0) | 0.927(0.0) | 0.927(0.0) | 0.927(0.0) | 0.926(0.0) | — | — | — | 0.927(0.0) | 0.0(0.0) | 1.0(1.0) |
| | R² ↑ | 0.853(0.001) | 0.852(0.002) | 0.852(0.003) | 0.853(0.001) | 0.851(0.001) | — | — | — | **0.854(0.0)** | 0.12(0.07) | 0.2(0.35) |

**Results:** all the testing results are summarized in Table 1. GAR outperforms all the competitors from the perspective of Pearson Correlation Coefficient and Spearman's Correlation Coefficient over all the tasks, which indicates the proposed pairwise losses can capture the rank or order based information better. GAR also enjoys competitive performance on MAE and RMSE evaluations. Finally, GAR achieves new SOTA results on AgeDB for both training from scratch and linear probe settings with ResNet18. For the scratch setting, GAR achieves 6.286(0.034) for MAE and 0.846(0.002) for $R^2$ score, while the previous SOTA is 6.40 for MAE and 0.830 for $R^2$ score; for the 2-stage linear probe on RNC pre-trained feature setting, GAR achieves 6.069(0.022) for MAE and 0.854(0.0) for $R^2$ score, while the previous SOTA is 6.14 for MAE and 0.850 for $R^2$ score (Zha et al., 2023).

### 4.3. Further Analysis

We provide practical running time comparison, ablation studies on different components, sensitivity analysis on $\alpha$ and batch size for GAR and difference with data standardization in this subsection.

**Running Time Comparison:** we run all compared methods sequentially on an exclusive cluster node with AMD EPYC 7402 24-Core Processor 2.0 GHz. Each method runs 1500 epochs for the five tasks on the tabular datasets. The results are summarized in Table 2. From the perspective of 'All Time', we find that GAR is as efficient as the most efficient conventional regression loss, such as MAE, MSE, etc. From the perspective of 'Loss Time', we find that GAR consistently takes slightly more time than conventional regression losses (but still in the same order/level). It is consistent with our theoretical result that GAR only takes linear time complexity. On the other hand, the pairwise latent feature based regression methods, such as RankSim, ConR and RNC, are clearly slower than GAR.

**Ablation Studies for GAR:** GAR constitutes three losses: $\mathcal{L}_c^{\text{MAE}}, \mathcal{L}_{\text{diff}}^{\text{MSE}}, \mathcal{L}_{\text{diffnorm}}^{p=2}$. We conduct ablation studies on different combination of the losses with the six real-world tasks on the tabular datasets. There are seven possible variants including GAR itself by different combinations of the three losses. Due to the limited space, we include the details in Appendix B.8. From the Tab. 5, we find that the proposed GAR outperforms the other six variants. Besides, the proposed pairwise losses, $\mathcal{L}_{\text{diff}}^{\text{MSE}}$ and $\mathcal{L}_{\text{diffnorm}}^{p=2}$, can also improve the conventional loss $\mathcal{L}_c^{\text{MAE}}$ even applied separately.

**Sensitivity on $\alpha$:** we conduct extensive hyper-parameter search for $\alpha$ in {0.1, 0.2, 0.25, 0.4, 0.5, 0.8, 1, 1.25, 2, 2.5, 4, 5, 10}. The initial learning rate and weight decay are still searched in the same previous sets that are described in Appendix B.6. We include the full results in Appendix B.9. From the box-plot, GAR is a little sensitive to $\alpha$ and it is suggested to tune it in the [0.1, 10] range.

**Sensitivity on Batch Sizes:** we additionally conduct ex-

*Table 2.* Running time (seconds/epoch) on five benchmark datasets. Repeated 1500 epochs, mean and standard deviation are reported. 'CCS': Concrete Compressive Strength. 'All Time': all the time consumed per epoch. 'Loss Time': the time for loss calculation and loss backward operation.

| Dataset | Concrete Compressive Strength | | Wine Quality | | Parkinson (Total) | | Super Conductivity | | IC50 | |
|---|---|---|---|---|---|---|---|---|---|---|
| Metric | All Time | Loss Time | All Time | Loss Time | All Time | Loss Time | All Time | Loss Time | All Time | Loss Time |
| MAE | 0.089(0.013) | 0.002(0.0) | 0.169(0.03) | 0.011(0.0) | 0.214(0.129) | 0.045(0.113) | 0.587(0.038) | 0.162(0.001) | 0.094(0.003) | 0.005(0.0) |
| MSE | 0.09(0.014) | 0.002(0.005) | 0.168(0.03) | 0.01(0.0) | 0.21(0.024) | 0.041(0.0) | 0.566(0.002) | 0.16(0.001) | 0.095(0.003) | 0.005(0.0) |
| Huber | 0.09(0.015) | 0.002(0.008) | 0.168(0.03) | 0.01(0.0) | 0.21(0.046) | 0.041(0.0) | 0.581(0.038) | 0.159(0.001) | 0.094(0.003) | 0.005(0.0) |
| Focal (MAE) | 0.09(0.013) | 0.002(0.001) | 0.177(0.03) | 0.016(0.0) | 0.214(0.024) | 0.046(0.0) | 0.603(0.037) | 0.178(0.001) | 0.096(0.041) | 0.005(0.0) |
| Focal (MSE) | 0.089(0.012) | 0.002(0.0) | 0.173(0.03) | 0.015(0.0) | 0.216(0.046) | 0.046(0.001) | 0.665(0.061) | 0.179(0.002) | 0.094(0.003) | 0.005(0.0) |
| RankSim | 0.364(0.046) | 0.275(0.045) | 0.224(0.021) | 0.068(0.002) | 1.982(0.033) | 1.81(0.03) | 6.876(1.024) | 6.428(0.977) | 0.362(0.009) | 0.271(0.006) |
| RNC | 0.371(0.012) | 0.275(0.01) | 2.178(0.016) | 2.014(0.015) | 2.195(0.029) | 2.02(0.029) | 7.989(0.231) | 7.545(0.168) | 0.275(0.004) | 0.178(0.003) |
| ConR | 0.103(0.011) | 0.014(0.004) | 0.269(0.026) | 0.108(0.001) | 0.307(0.023) | 0.138(0.001) | 0.943(0.057) | 0.508(0.022) | 0.108(0.002) | 0.02(0.0) |
| GAR (ours) | 0.086(0.012) | 0.003(0.0) | 0.177(0.026) | 0.026(0.0) | 0.22(0.024) | 0.056(0.001) | 0.64(0.039) | 0.213(0.002) | 0.09(0.003) | 0.006(0.0) |

periments on different batch sizes, {32, 64, 128, 256, 512}, for GAR on Concrete Compressive Strength, Parkinson (Total) and Parkinson (Motor) datasets. Details and results are included in Appendix B.10. GAR is also a little sensitive to batch size and suggested to be tuned in {32, 64, 128} in practice.

**Difference with Data Standardization:** to emphasize the difference between Eq. 5, 7 and data centering or standardization, we include extensive discussions and experiments in the Appendix B.11 and Tab. 7. The proposed GAR enjoys similar superior performance on the datasets after data standardization as the main results in Tab. 1.

## 5. Limitations

The scope for this work is limited to improving learning label pairwise difference for regression on clean data setting, without considering noises, outliers, or distributional shifts. Based on the Theorem 4, we relate GAR to capture the gradient of the ground truth function. However, GAR might be misled if the training data does not reflect the underlying ground truth function. It would be an interesting future work to research the robustness for GAR under extensive settings.

## 6. Conclusion

In this work, we propose GAR (Gradient Aligned Regression) via two novel pairwise losses for regression. Similar to the prior research, the proposed losses can help model preserve rank information from ground truths better. Different with the prior studies, the proposed pairwise losses directly capture the pairwise label difference in the label space without any approximation, which additionally allows us to make transformations and reduce the quadratic complexity to linear complexity. We proved that the two proposed pairwise losses are equivalent to the variance of prediction errors and the negative Pearson correlation coefficient between the predictions and ground truths. Last but not least, we provide theoretical insights for learning the pairwise label difference to capturing the gradients of the

ground truth function. We demonstrate the effectiveness of GAR on two synthetic datasets and eight real-world tasks on five tabular datasets and one image dataset. Running time experiments are conducted to support that GAR is as efficient as the conventional regression loss. Experimental ablation studies are conducted to demonstrate the effectiveness separately for the two proposed pairwise losses.

## Acknowledgement

This research was partially funded by NIH U01HG012069 and Allen Distinguished Investigator Award, a Paul G. Allen Frontiers Group advised grant of the Paul G. Allen Family Foundation.

## Impact Statement

This paper presents work whose goal is to advance the field of Machine Learning. There are many potential societal consequences of our work, none which we feel must be specifically highlighted here.

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

## A. Proofs

### A.1. Proof for Theorem 1

**Proof**

Recall that $\bar{\delta}_{\mathbf{x}}^f = \frac{1}{N} \sum_{i=1}^{N} \delta_{\mathbf{x}_i}^f$ denotes the empirical mean.

$$
\begin{aligned}
\mathcal{L}_{\text{diff}}^{\text{MSE}} &= \frac{1}{N^2} \sum_{i=1}^{N} \sum_{j=1}^{N} \frac{1}{2} \big[ f(\mathbf{x}_i) - \mathcal{Y}(\mathbf{x}_i) - \big( f(\mathbf{x}_j) - \mathcal{Y}(\mathbf{x}_j) \big) \big]^2 \\
&= \frac{1}{2N^2} \sum_{i=1}^{N} \sum_{j=1}^{N} (\delta_{\mathbf{x}_i}^f - \delta_{\mathbf{x}_j}^f)^2 \\
&= \frac{1}{2N^2} \sum_{i=1}^{N} \sum_{j=1}^{N} (\delta_{\mathbf{x}_i}^f - \bar{\delta}_{\mathbf{x}}^f + \bar{\delta}_{\mathbf{x}}^f - \delta_{\mathbf{x}_j}^f)^2 \\
&= \frac{1}{N} \sum_{i=1}^{N} (\delta_{\mathbf{x}_i}^f - \bar{\delta}_{\mathbf{x}}^f)^2 \\
&= \text{Var}(\delta_{\mathbf{x}}^f),
\end{aligned}
$$

where the cross terms expanded out in the third equation can be canceled when iterate across all data samples.

∎

### A.2. Proof for Corollary 3

**Proof**

Recall that $df_{i,j} = f(\mathbf{x}_i) - f(\mathbf{x}_j)$ and $d\mathcal{Y}_{i,j} = \mathcal{Y}(\mathbf{x}_i) - \mathcal{Y}(\mathbf{x}_j)$, and $\mathbf{df} = [df_{1,1}, df_{1,2}, ..., df_{N,N-1}, df_{N,N}]$, $\mathbf{d\mathcal{Y}} = [d\mathcal{Y}_{1,1}, d\mathcal{Y}_{1,2}, ..., d\mathcal{Y}_{N,N-1}, d\mathcal{Y}_{N,N}]$.

$$
\begin{aligned}
\mathcal{L}_{\text{diffnorm}}^{p=2} &= \| \frac{\mathbf{df}}{\|\mathbf{df}\|_2} - \frac{\mathbf{d\mathcal{Y}}}{\|\mathbf{d\mathcal{Y}}\|_2} \|_2^2 \\
&\quad \text{(by Theorem 1)} \\
&= \| \frac{\mathbf{df}}{\sqrt{2N^2 \text{Var}(f)}} - \frac{\mathbf{d\mathcal{Y}}}{\sqrt{2N^2 \text{Var}(y)}} \|_2^2 \\
&= \frac{1}{2N^2} \sum_{i=1}^{N} \sum_{j=1}^{N} \left( \frac{f(\mathbf{x}_i) - f(\mathbf{x}_j)}{\text{Var}(f)} - \frac{\mathcal{Y}(\mathbf{x}_i) - \mathcal{Y}(\mathbf{x}_j)}{\text{Var}(y)} \right)^2 \\
&= \frac{1}{2N^2} \sum_{i=1}^{N} \sum_{j=1}^{N} \left( \frac{f(\mathbf{x}_i) - \bar{f} + \bar{f} - f(\mathbf{x}_j)}{\text{Var}(f)} - \frac{\mathcal{Y}(\mathbf{x}_i) - \bar{y} + \bar{y} - \mathcal{Y}(\mathbf{x}_j)}{\text{Var}(y)} \right)^2 \\
&\quad \text{(by algebras, there are 2N repeats)} \\
&= \frac{2N}{2N^2} \left[ \sum_{i=1}^{N} (\frac{f(\mathbf{x}_i) - \bar{f}}{\text{Var}(f)})^2 - 2(\frac{f(\mathbf{x}_i) - \bar{f}}{\text{Var}(f)})(\frac{\mathcal{Y}(\mathbf{x}_i) - \bar{y}}{\text{Var}(y)}) + (\frac{\mathcal{Y}(\mathbf{x}_i) - \bar{y}}{\text{Var}(y)})^2 \right] \\
&= \frac{1}{2N} \sum_{i=1}^{N} \left( \frac{(f(\mathbf{x}_i) - \bar{f})}{\sqrt{\text{Var}(f)}} - \frac{(y_i - \bar{y})}{\sqrt{\text{Var}(y)}} \right)^2 \\
&= 1 - \frac{\text{Cov}(f, y)}{\sqrt{\text{Var}(f)\text{Var}(y)}} = 1 - \rho(f, y).
\end{aligned}
$$

∎

### A.3. Proof for Theorem 4

**Proof**

$f(\mathbf{x}_1) - f(\mathbf{x}_2) = \mathcal{Y}(\mathbf{x}_1) - \mathcal{Y}(\mathbf{x}_2), \ \forall \big(\mathbf{x}_1, \mathcal{Y}(\mathbf{x}_1)\big), \big(\mathbf{x}_2, \mathcal{Y}(\mathbf{x}_2)\big) \in \mathcal{D} \implies \nabla^k f(\mathbf{x}) = \nabla^k \mathcal{Y}(\mathbf{x}), \ \forall (\mathbf{x}, \mathcal{Y}(\mathbf{x})) \in \mathcal{D}, \ k = \{1, 2, ...\}$.

First, we prove that $\nabla f(\mathbf{x}) = \nabla \mathcal{Y}(\mathbf{x}), \ \forall (\mathbf{x}, \mathcal{Y}(\mathbf{x})) \in \mathcal{D}$.

Take two neighbors $\mathbf{x}_1, \mathbf{x}_2$ that are infinitely close to an arbitrary $\mathbf{x}$. By mean value theorem for multiple variables (Rudin et al., 1976): $f(\mathbf{x}_1) - f(\mathbf{x}_2) = \nabla f(\xi)^\top(\mathbf{x}_1 - \mathbf{x}_2)$, $\mathcal{Y}(\mathbf{x}_1) - \mathcal{Y}(\mathbf{x}_2) = \nabla \mathcal{Y}(\gamma)^\top(\mathbf{x}_1 - \mathbf{x}_2)$, where $\xi, \gamma$ are two points that are infinitely close to $\mathbf{x}$. Because $f(\mathbf{x}_1) - f(\mathbf{x}_2) = \mathcal{Y}(\mathbf{x}_1) - \mathcal{Y}(\mathbf{x}_2), \ \forall \big(\mathbf{x}_1, \mathcal{Y}(\mathbf{x}_1)\big), \big(\mathbf{x}_2, \mathcal{Y}(\mathbf{x}_2)\big) \in \mathcal{D}$, we have $\nabla f(\xi)^\top(\mathbf{x}_1 - \mathbf{x}_2) = \nabla \mathcal{Y}(\gamma)^\top(\mathbf{x}_1 - \mathbf{x}_2)$.

As $\mathbf{x}_1, \mathbf{x}_2$ are infinitely close, $\xi, \gamma$ collapse to the same point $\mathbf{x}$. Once $\mathbf{x}_1, \mathbf{x}_2$ are selected, define $\mathbf{x}_1 - \mathbf{x}_2 = \epsilon \cdot \mathbf{a}$, where $\mathbf{a}$ is a unit length vector and we will decrease the scalar $\epsilon$ to scale down the vector $\mathbf{x}_1 - \mathbf{x}_2$. As a consequence, $\lim_{\mathbf{x}_1 \to \mathbf{x}, \mathbf{x}_2 \to \mathbf{x}} \nabla f(\xi)^\top(\mathbf{x}_1 - \mathbf{x}_2) = \nabla f(\mathbf{x})^\top(\lim_{\epsilon \to 0} \epsilon \cdot \mathbf{a})$; $\lim_{\mathbf{x}_1 \to \mathbf{x}, \mathbf{x}_2 \to \mathbf{x}} \nabla \mathcal{Y}(\gamma)^\top(\mathbf{x}_1 - \mathbf{x}_2) = \nabla \mathcal{Y}(\mathbf{x})^\top(\lim_{\epsilon \to 0} \epsilon \cdot \mathbf{a})$. Plug them back to the left and right side of the equation: $\nabla f(\mathbf{x})^\top(\lim_{\epsilon \to 0} \epsilon \cdot \mathbf{a}) = \nabla \mathcal{Y}(\mathbf{x})^\top(\lim_{\epsilon \to 0} \epsilon \cdot \mathbf{a})$, by L'Hôpital's rule $\nabla f(\mathbf{x})^\top \cdot \mathbf{a} = \nabla \mathcal{Y}(\mathbf{x})^\top \cdot \mathbf{a} \cdot \frac{\lim_{\epsilon \to 0} \epsilon}{\lim_{\epsilon \to 0} \epsilon} = \nabla \mathcal{Y}(\mathbf{x})^\top \cdot \mathbf{a}$. Hence, $\nabla f(\mathbf{x}) = \nabla \mathcal{Y}(\mathbf{x})$ because $\mathbf{a}$ can be any direction constructed by $\mathbf{x}_1, \mathbf{x}_2$. Therefore, $\nabla f(\mathbf{x}) = \nabla \mathcal{Y}(\mathbf{x})$.

Next, we show $\nabla^k f(\mathbf{x}) = \nabla^k \mathcal{Y}(\mathbf{x}), \ \forall (\mathbf{x}, \mathcal{Y}(\mathbf{x})) \in \mathcal{D}, \ k = \{2, ...\}$ by induction. Assume we have $\nabla^k f(\mathbf{x}) = \nabla^k \mathcal{Y}(\mathbf{x}), \ \forall (\mathbf{x}, \mathcal{Y}(\mathbf{x})) \in \mathcal{D}$, to show $\nabla^{k+1} f(\mathbf{x}) = \nabla^{k+1} \mathcal{Y}(\mathbf{x}), \ \forall (\mathbf{x}, \mathcal{Y}(\mathbf{x})) \in \mathcal{D}$.

Denote i-th element from $\nabla^k f(\mathbf{x})$ as $[\nabla^k f(\mathbf{x})]_i$. Take two neighbors $\mathbf{x}_1, \mathbf{x}_2$ that are infinitely close to arbitrary $\mathbf{x}$ and apply mean value theorem for multiple variables solely on the i-th element: $[\nabla^k f(\mathbf{x}_1)]_i - [\nabla^k f(\mathbf{x}_2)]_i = \nabla_i^{k+1} f(\xi_i)(\mathbf{x}_1 - \mathbf{x}_2)$, $[\nabla^k \mathcal{Y}(\mathbf{x}_1)]_i - [\nabla^k \mathcal{Y}(\mathbf{x}_2)]_i = \nabla_i^{k+1} \mathcal{Y}(\gamma_i)(\mathbf{x}_1 - \mathbf{x}_2)$, where $\nabla_i^{k+1} f(\xi_i)$ represents the higher order gradient taken for $[\nabla^k f(\mathbf{x})]_i$ regarding to variables. It is worth noting that $\xi_i$ can be different points for different $i$, which hamper the general mean value theorem for vector-valued functions in the literature (Rudin et al., 1976) but doesn't harm the specific proof here. Similarly, because $[\nabla^k f(\mathbf{x}_1)]_i - [\nabla^k f(\mathbf{x}_2)]_i = [\nabla^k \mathcal{Y}(\mathbf{x}_1)]_i - [\nabla^k \mathcal{Y}(\mathbf{x}_2)]_i, \ \forall \mathbf{x}_1, \mathbf{x}_2$, we have $\nabla_i^{k+1} f(\xi_i)(\mathbf{x}_1 - \mathbf{x}_2) = \nabla_i^{k+1} \mathcal{Y}(\gamma_i)(\mathbf{x}_1 - \mathbf{x}_2)$.

As $\mathbf{x}_1, \mathbf{x}_2$ are infinitely close, $\xi_i, \gamma_i$ collapse to the same point $\mathbf{x}, \ \forall i$. Utilize the same logic with L'Hôpital's rule in the previous first order gradient case, we can cancel the effects of the magnitude for $\mathbf{x}_1 - \mathbf{x}_2$ closing to 0 but preserve the direction information. Therefore, $\nabla_i^{k+1} f(\mathbf{x}) = \nabla_i^{k+1} \mathcal{Y}(\mathbf{x}), \ \forall i$; thereby $\nabla^{k+1} f(\mathbf{x}) = \nabla^{k+1} \mathcal{Y}(\mathbf{x})$.

Next, we prove the other direction.

$\nabla^k f(\mathbf{x}) = \nabla^k \mathcal{Y}(\mathbf{x}), \ \forall (\mathbf{x}, \mathcal{Y}(\mathbf{x})) \in \mathcal{D}, \ k = \{1, 2, ...\} \implies f(\mathbf{x}_1) - f(\mathbf{x}_2) = \mathcal{Y}(\mathbf{x}_1) - \mathcal{Y}(\mathbf{x}_2), \ \forall \big(\mathbf{x}_1, \mathcal{Y}(\mathbf{x}_1)\big), \big(\mathbf{x}_2, \mathcal{Y}(\mathbf{x}_2)\big) \in \mathcal{D}$.

$\nabla^k f(\mathbf{x}) = \nabla^k \mathcal{Y}(\mathbf{x}), \ \forall (\mathbf{x}, \mathcal{Y}(\mathbf{x})) \in \mathcal{D} \implies \nabla f(\mathbf{x}) = \nabla \mathcal{Y}(\mathbf{x}), \ \forall (\mathbf{x}, \mathcal{Y}(\mathbf{x}))$. We take infinitely small but infinitely many steps $\epsilon_i, i = 1, ..., \infty$ that constitute a path from $\mathbf{x}_1$ to $\mathbf{x}_2$. By mean value theorem for multiple variables, $f(\mathbf{x}_1 + \sum_{i=1}^k \epsilon_i) - f(\mathbf{x}_1 + \sum_{i=1}^{k-1} \epsilon_i) = \nabla f(\mathbf{x}_1 + \sum_{i=1}^k \epsilon_i)^\top(\epsilon_k), \quad \mathcal{Y}(\mathbf{x}_1 + \sum_{i=1}^k \epsilon_i) - \mathcal{Y}(\mathbf{x}_1 + \sum_{i=1}^{k-1} \epsilon_i) = \nabla \mathcal{Y}(\mathbf{x}_1 + \sum_{i=1}^k \epsilon_i)^\top(\epsilon_k)$.

Therefore, $\sum_{k=1}^\infty f(\mathbf{x}_1 + \sum_{i=1}^k \epsilon_i) - f(\mathbf{x}_1 + \sum_{i=1}^{k-1} \epsilon_i) = \sum_{k=1}^\infty \mathcal{Y}(\mathbf{x}_1 + \sum_{i=1}^k \epsilon_i) - \mathcal{Y}(\mathbf{x}_1 + \sum_{i=1}^{k-1} \epsilon_i)$; and by telescoping sum, $f(\mathbf{x}_2) - f(\mathbf{x}_1) = \mathcal{Y}(\mathbf{x}_2) - \mathcal{Y}(\mathbf{x}_1)$.

∎

### A.4. Proof for Theorem 6

**Proof**

$$\mathcal{L}_{\text{GAR}}^{\text{KL}}(\mathcal{L}_1, ..., \mathcal{L}_M; \alpha) = \max_{\mathbf{p} \in \Delta_M} \sum_{i=1}^{M} p_i \log \mathcal{L}_i - \alpha \text{KL}(\mathbf{p} | \frac{\mathbf{1}}{M}).$$

By Lagrangian multiplier:

$$\mathcal{L}_{\text{GAR}}^{\text{KL}}(\mathcal{L}_1, ..., \mathcal{L}_M; \alpha) = \max_{\mathbf{p}} \min_{\mathbf{a} \geq 0, b} \sum_{i=1}^{M} p_i \log \mathcal{L}_i - \alpha \text{KL}(\mathbf{p} | \frac{\mathbf{1}}{M}) + \sum_{i=1}^{M} a_i p_i - b(\sum_{i=1}^{M} p_i - 1). \tag{12}$$

By stationary condition of $\mathbf{p}$:

$$p_i^* = \frac{1}{M} \exp \big( \frac{\log \mathcal{L}_i + a_i - b}{\alpha} - 1 \big).$$

Take the optimal $\mathbf{p}^*$ back to E.q. 12:

$$\mathcal{L}_{\text{GAR}}^{\text{KL}}(\mathcal{L}_1, ..., \mathcal{L}_M; \alpha) = \min_{\mathbf{a} \geq 0, b} b + \alpha \sum_{i=1}^{M} \frac{1}{M} \exp \big( \frac{\log \mathcal{L}_i + a_i - b}{\alpha} - 1 \big). \tag{13}$$

$a_i$ can be minimized as 0, $\forall i$, because $\alpha \geq 0$. Further take the stationary condition for $b$:

$$1 = \sum_{i=1}^{M} \frac{1}{M} \exp \big( \frac{\log \mathcal{L}_i - b^*}{\alpha} - 1 \big)$$

$$\exp(b^*/\alpha) = \sum_{i=1}^{M} \frac{1}{M} \exp \big( \frac{\log \mathcal{L}_i}{\alpha} - 1 \big).$$

Take the optimal solution $b^*$ back to E.q. 13:

$$\mathcal{L}_{\text{GAR}}^{\text{KL}}(\mathcal{L}_1, ..., \mathcal{L}_M; \alpha) = \alpha \log \left[ \sum_{i=1}^{M} \frac{1}{M} \exp \big( \frac{\log \mathcal{L}_i}{\alpha} - 1 \big) \right] + \alpha \tag{14}$$

$$= \alpha \log \sum_{i=1}^{M} \frac{1}{M} \exp \big( \frac{\log \mathcal{L}_i}{\alpha} \big)$$

$$= \alpha \log(\frac{1}{M} \sum_{i=1}^{M} \mathcal{L}_i^{1/\alpha}).$$

Therefore,

$$\exp \big( \mathcal{L}_{\text{GAR}}^{\text{KL}}(\mathcal{L}_1, ..., \mathcal{L}_M; \alpha) \big) = (\frac{1}{M} \sum_{i=1}^{M} \mathcal{L}_i^{1/\alpha})^{\alpha}.$$

When $\alpha = 1$, we recover the Arithmetic Mean. Next we discuss the cases that $\alpha \to +\infty$ and $\alpha \to 0$. By L'Hôpital's rule:

$$\lim_{\alpha \to +\infty} \mathcal{L}_{\text{GAR}}^{\text{KL}}(\mathcal{L}_1, ..., \mathcal{L}_M; \alpha) = \lim_{\alpha \to +\infty} \frac{\log(\frac{1}{M} \sum_{i=1}^{M} \mathcal{L}_i^{1/\alpha})}{1/\alpha}$$

$$= \lim_{\alpha \to +\infty} \frac{\sum_{i=1}^{M} \mathcal{L}_i^{1/\alpha} \log \mathcal{L}_i}{\mathcal{L}_i^{1/\alpha}}$$

$$= \frac{\sum_{i=1}^{M} \log \mathcal{L}_i}{M}.$$

Hence, $\lim_{\alpha \to +\infty} \exp \big( \mathcal{L}_{\text{GAR}}^{\text{KL}} \big) = (\Pi_{i=1}^{M} \mathcal{L}_i)^{\frac{1}{M}}$. By the same logic, $\lim_{\alpha \to 0} \exp \big( \mathcal{L}_{\text{GAR}}^{\text{KL}} \big) = \max_{i=1}^{M} \mathcal{L}_i$.

∎

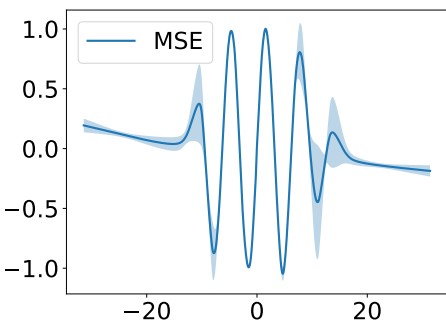 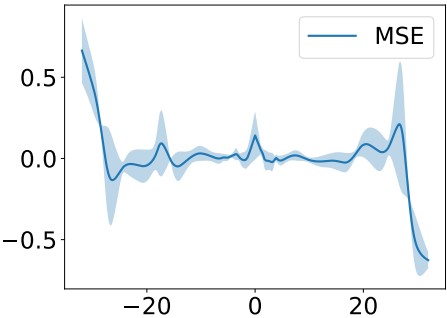

*Figure 3.* MSE predictions on synthetic datasets.

# B. More Experiments

## B.1. Dataset Statistics

The statistics for benchmark datasets are summarized as following. The AgeDB is a image dataset where the image size is not identical but with fixed center crop size. The # of feature is measured after data pre-processing as model input.

*Table 3.* Dataset Statistics

| Dataset | # of instance | # of feature | # of targets |
|---|---|---|---|
| Concrete Compressive Strength | 1030 | 8 | 1 |
| Wine Quality | 4898 | 11 | 1 |
| Parkinson | 5875 | 19 | 2 |
| Super Conductivity | 21263 | 81 | 1 |
| IC50 | 429 | 966 | 15 |
| AgeDB | 16488 | 224×224 | 1 |

## B.2. Data Pre-processing for IC50

For IC50 (Garnett et al., 2012), there are some missing values for all the 130 drugs across all tumour cell lines, which make the model training and evaluation inconvenient. We filter out the drugs with more than 28% missing values, which gives us targets for 15 drugs. Then, we select the data samples (cell lines) that don't have any missing values for the 15 drugs, which end up with 429 samples. We utilize dummy variables to encode data categorical features (tissue type, cancer type, genetic information).

## B.3. MSE on Synthetic Datasets

Due to the limited space and high similarity between MAE and MSE, we include the predictions for MSE in Fig. 3. As mentioned in the main content and Fig. 1, 2, MSE is similar with MAE, which performs worse on capturing complex data rank pattern.

## B.4. Symbolic Regression on the Synthetic Datasets

We adopt PySR (Cranmer, 2023) on the synthetic experiments. We present the results for without trigonometric functions and with trigonometric functions prior knowledge in Fig 4 and Fig. 5 for Sine and Squared Sine datasets. The predictions (dashed line) is the mean prediction values based on 5 independent runs of PySR.

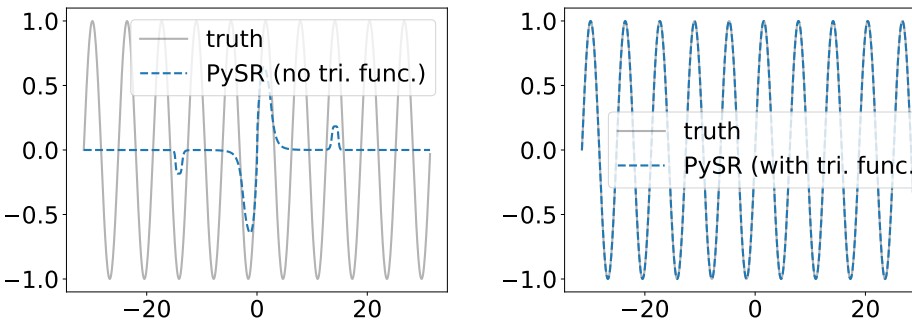

*Figure 4.* PySR on Sine dataset.

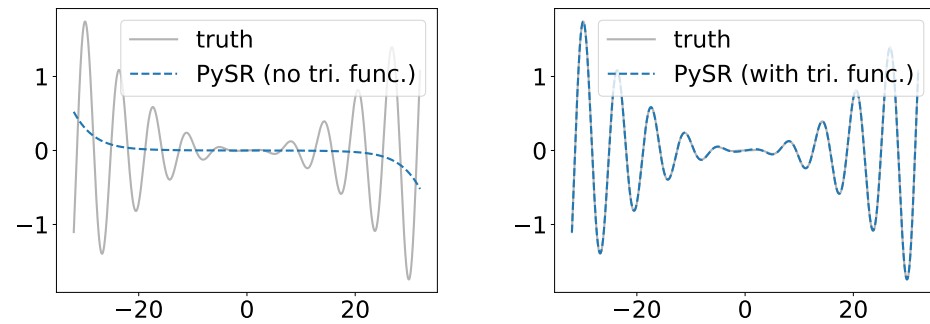

*Figure 5.* PySR on Squared Sine dataset.

## B.5. Backbone Model Complexity Variation on Sine Dataset

To demonstrate the effectiveness for the proposed GAR. We varies the backbone model complexity from {2,4,6,8,10} hidden layers for the previous FFNN backbone model, besides of the previous 5 hidden layers FFNN results in the main content. $\alpha$ for GAR is tuned in {0.5,1,2}. All of other settings are kept the same as previously described in the main content. The results are presented in the Fig 6. Based on the results, we have two observations: 1) increasing model complexity can generally improve the learning performance but may saturate at certain level (e.g. for MAE, 8 hidden layers model is better than 10 hidden layers model; for MSE, 6 hidden layers model is better 8 and 10 hidden layers model). 2) GAR consistently demonstrates its superiority over other baseline methods for comparison.

## B.6. Experimental Setting on Real World Datasets

For tabular datasets, we uniformly randomly split 20% data as testing; the remaining 80% as training and validation, where we conduct 5-fold-cross-validation with random seed set as 123. The total training epochs is set as 100 and batch size is set as 256. The weight decay for each method is tuned in {1e-3, 1e-4, 1e-5}; we utilize SGD with momentum (set as 0.9) optimizer and tune the initial learning rate for baseline method in {1e-1, 1e-2, 1e-3, 1e-4, 1e-5}, which is stage-wised decreased by 10 folds at the end of 50-th and 75-th epoch. The switching hyper-parameter $\delta$ for Huber loss and the scaling hyper-parameter $\beta$ for Focal (MAE) or Focal (MSE) loss are tuned in {0.25, 1, 4}. The interpolation hyper-parameter $\lambda$ for RankSim is tuned in {0.5, 1, 2} and the balancing hyper-parameter $\gamma$ is fixed as 100 as suggested by their sensitivity study in their Appendix C.4 (Gong et al., 2022). The temperature hyper-parameter for RNC is tuned in {1,2,4}; the first 50 epochs are used for RNC pre-training and the remaining 50 epochs are used for fine-tuning with MAE loss. The linear combination hyper-parameters $\alpha$ is fixed as 1, $\beta$ is tuned in {0.2, 1, 4} for ConR, as suggested by the ablation studies in their Appendix A.5 (Keramati et al., 2023). The robust reconciliation hyper-parameter $\alpha$ for GAR is tuned in {0.1, 1, 10}. The model performance is evaluated with the following 4 metrics: MAE, RMSE, Pearson correlation coefficient,

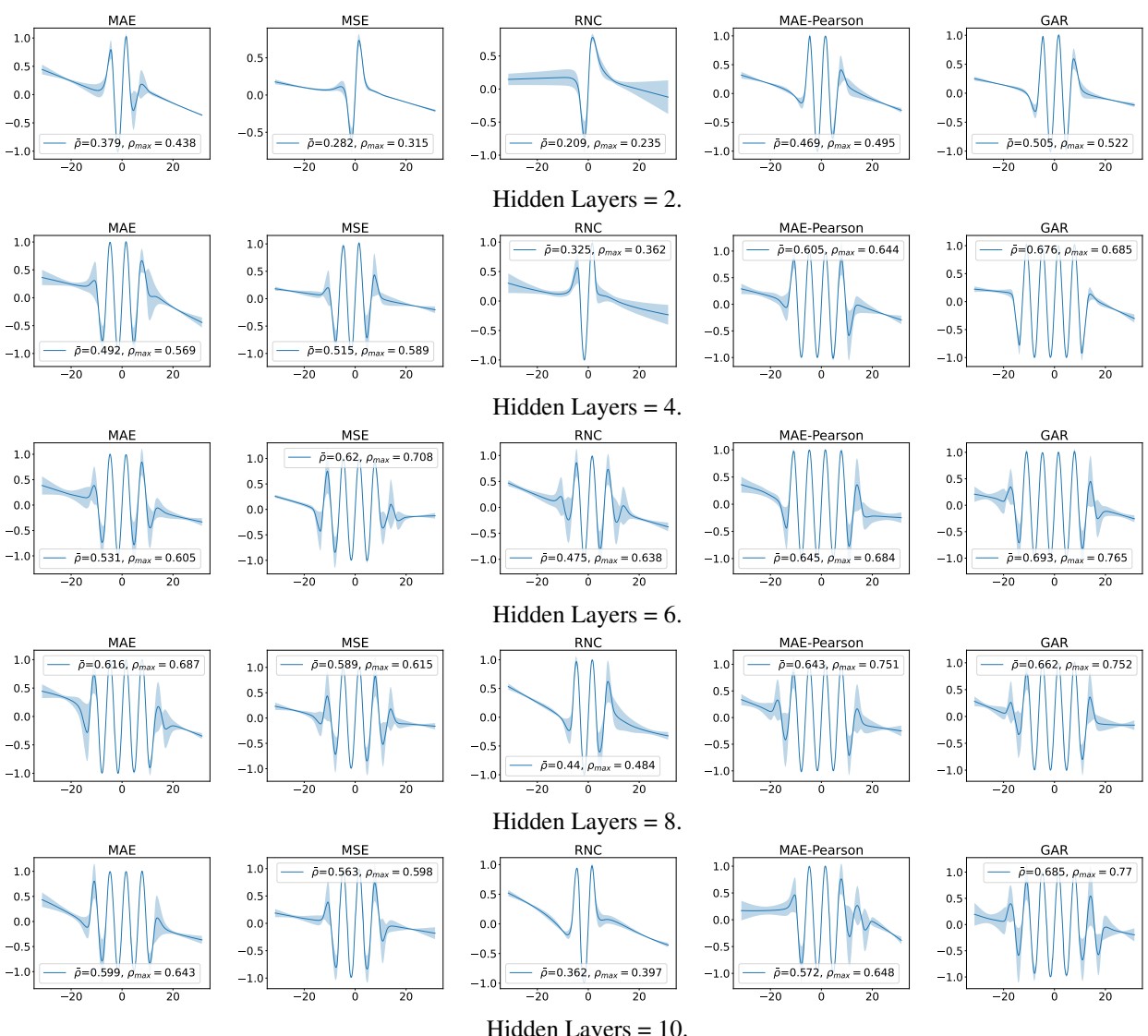

Figure 6. Predictions from different numbers of hidden layers for FFNN on Sine dataset. $\bar{\rho}$ stands for the averaged Pearson correlation coefficient with the ground truth over the 5 trials; $\rho_{\max}$ is the maximal Pearson correlation coefficient with the ground truth over the 5 trials.

Spearman's rank correlation coefficient. **The best testing performance for each baseline method on each training fold is decided by its best validation performance on the same training fold over (initial learning rate, weight decay, method-special-hyper-parameter, the epoch for evaluation) combinations.** For IC50 with 15 targets, we report the averaged value across all targets. The selected $\alpha$ for different datasets on different data fold are summarized in Tab. 4. For AgeDB dataset, we follow the same setting for RNC linear probe task and similar setting for training from scratch task. We include the complete details in Appendix B.7. Because RNC is a pre-training method, it doesn't have evaluation on training from scratch task. Instead, all baselines in linear probe are based on RNC pre-trained latent features. Because RankSim and ConR are proposed to manipulate latent feature, they don't have evaluation on linear probe task where the trainable model is linear. We also include the coefficient of determination ($R^2$) as an evaluation metric for AgeDB in addition to the previous four evaluation metrics, to compare with prior research.

*Table 4.* The selected $\alpha$ on the tabular data respecting to different evaluation metrics. Standard deviations are included in the parenthesis.

| Dataset | Data Fold | #1 | #2 | #3 | #4 | #5 | Average |
|---------|-----------|-----|-----|-----|-----|-----|---------|
| Concrete Compressive Strength | MAE | 10 | 10 | 10 | 10 | 10 | 10.0(0.0) |
| | RMSE | 10 | 1 | 10 | 10 | 10 | 8.2(3.6) |
| | Pearson | 10 | 1 | 10 | 10 | 10 | 8.2(3.6) |
| | Spearman | 10 | 1 | 10 | 10 | 10 | 8.2(3.6) |
| Wine Quality | MAE | 0.1 | 0.1 | 0.1 | 0.1 | 0.1 | 0.1(0.0) |
| | RMSE | 10 | 10 | 10 | 10 | 10 | 10.0(0.0) |
| | Pearson | 10 | 10 | 10 | 10 | 10 | 10.0(0.0) |
| | Spearman | 10 | 10 | 10 | 10 | 10 | 10.0(0.0) |
| Parkinson (Total) | MAE | 10 | 1 | 10 | 1 | 10 | 6.4(4.409) |
| | RMSE | 0.1 | 0.1 | 0.1 | 10 | 0.1 | 2.08(3.96) |
| | Pearson | 0.1 | 0.1 | 0.1 | 10 | 0.1 | 2.08(3.96) |
| | Spearman | 0.1 | 0.1 | 0.1 | 10 | 0.1 | 2.08(3.96) |
| Parkinson (Motor) | MAE | 0.1 | 0.1 | 0.1 | 0.1 | 0.1 | 0.1(0.0) |
| | RMSE | 0.1 | 0.1 | 0.1 | 0.1 | 0.1 | 0.1(0.0) |
| | Pearson | 0.1 | 0.1 | 0.1 | 0.1 | 0.1 | 0.1(0.0) |
| | Spearman | 0.1 | 0.1 | 0.1 | 0.1 | 0.1 | 0.1(0.0) |
| SuperConductivity | MAE | 10 | 10 | 1 | 10 | 10 | 8.2(3.6) |
| | RMSE | 10 | 10 | 1 | 1 | 1 | 4.6(4.409) |
| | Pearson | 0.1 | 10 | 1 | 1 | 0.1 | 2.44(3.801) |
| | Spearman | 10 | 10 | 10 | 10 | 10 | 10.0(0.0) |
| IC50 | MAE | 1 | 1 | 1 | 1 | 1 | 1.0(0.0) |
| | RMSE | 1 | 1 | 1 | 0.1 | 0.1 | 0.64(0.441) |
| | Pearson | 1 | 1 | 10 | 10 | 10 | 6.4(4.409) |
| | Spearman | 1 | 1 | 10 | 10 | 10 | 6.4(4.409) |

## B.7. Experiment Settings for AgeDB

We follow the the same experiment settings from the previous work (Zha et al., 2023). The age range is between 0 and 101. It is split into 12208/2140/2140 images for training/validation/testing sets. The SGD optimizer and cosine learning rate annealing is utilized for training (Loshchilov & Hutter, 2016). The batch size is set to 256. For both training from scratch and linear probe experiments, we select the best learning rates and weight decays for each dataset by grid search, with a grid of learning rates from {0.01, 0.05, 0.1, 0.2, 0.5, 1.0} and weight decays from {1e-6, 1e-5, 1e-4, 1e-3}. For the predictor training of two-stage methods, we adopt the same search setting as above except for adding additional 0 weight decay to the search choices of weight decays. The RNC temperature parameter is search from {0.1, 0.2, 0.5, 1.0, 2.0, 5.0} and select the best, which is 2.0. We train all one-stage methods and the encoder of two-stage methods for 400 epochs, and the linear regressor of two-stage methods for 100 epochs. The hyper-parameters for all the compared methods are searched in the same region as described in appendix B.6. For training from scratch experiments, because the ConR (Keramati et al., 2023) is proposed to use two data augmentations, we utilize two data augmentations for each training data for all compared methods. The $\alpha$ for GAR is selected as 0.1 for training from scratch setting and as 1.0 for linear probe setting. For each method, we randomly repeat five times with different random seeds to report the mean and standard deviation for evaluations.

## B.8. Ablation Studies on GAR

Recall that GAR constitutes of 3 losses: $\mathcal{L}_c^{\text{MAE}}$, $\mathcal{L}_{\text{diff}}^{\text{MSE}}$ and $\mathcal{L}_{\text{diffnorm}}^{p=2}$. Therefore, we conduct the similar experiments as described in Appendix B.6 for the extra six variants of GAR on the tabular datasets as the ablation studies. The results are summarized in Tab. 5. From the results, we observe that the proposed GAR with all the three components performs the best over all the variants. Besides, the proposed pairwise losses, $\mathcal{L}_{\text{diff}}^{\text{MSE}}$ and $\mathcal{L}_{\text{diffnorm}}^{p=2}$, can also improve the conventional loss $\mathcal{L}_c^{\text{MAE}}$ even applied separately.

*Table 5.* GAR constitutes of 3 different losses; thereby there are 6 variants for ablation studies on GAR by taking off the components. ↓ means the smaller the better; ↑ means the larger the better. Mean and standard deviation (in the parenthesis) values are reported. The overall ranks for each variant are reported in the last row.

| Dataset | Method | $\mathcal{L}_c^{\text{MAE}}$ | $\mathcal{L}_{\text{diff}}^{\text{MSE}}$ | $\mathcal{L}_{\text{diffnorm}}^{p=2}$ | $\mathcal{L}_c^{\text{MAE}}$ and $\mathcal{L}_{\text{diff}}^{\text{MSE}}$ | $\mathcal{L}_c^{\text{MAE}}$ and $\mathcal{L}_{\text{diffnorm}}^{p=2}$ | $\mathcal{L}_{\text{diff}}^{\text{MSE}}$ and $\mathcal{L}_{\text{diffnorm}}^{p=2}$ | GAR |
|---|---|---|---|---|---|---|---|---|
| Concrete Compressive Strength | MAE ↓ | 5.242(0.212) | 22.518(6.911) | 27.182(4.86) | 4.917(0.092) | 4.621(0.137) | 19.104(6.491) | **4.603(0.075)** |
| | RMSE ↓ | 7.188(0.369) | 23.632(6.829) | 29.125(4.202) | 6.502(0.21) | 6.244(0.138) | 20.266(6.088) | **6.222(0.148)** |
| | Pearson ↑ | 0.903(0.01) | 0.92(0.003) | 0.928(0.005) | 0.922(0.003) | 0.928(0.003) | 0.925(0.005) | **0.929(0.004)** |
| | Spearman ↑ | 0.903(0.006) | 0.925(0.003) | **0.935(0.005)** | 0.928(0.002) | 0.928(0.005) | 0.929(0.005) | 0.931(0.003) |
| Wine Quality | MAE ↓ | **0.494(0.004)** | 5.334(0.241) | 4.386(1.165) | 0.498(0.006) | 0.496(0.006) | 4.278(1.875) | **0.494(0.004)** |
| | RMSE ↓ | 0.711(0.004) | 5.381(0.238) | 4.791(1.053) | 0.694(0.005) | 0.691(0.003) | 4.347(1.83) | **0.69(0.004)** |
| | Pearson ↑ | 0.582(0.003) | **0.622(0.008)** | 0.621(0.004) | 0.607(0.009) | 0.608(0.013) | 0.615(0.003) | 0.613(0.011) |
| | Spearman ↑ | 0.612(0.008) | 0.637(0.006) | 0.636(0.005) | 0.636(0.006) | 0.631(0.01) | **0.638(0.005)** | 0.635(0.007) |
| Parkinson (Total) | MAE ↓ | 3.358(0.045) | 27.563(0.902) | 24.393(2.48) | 2.746(0.068) | 2.648(0.13) | 26.861(0.707) | **2.64(0.071)** |
| | RMSE ↓ | 5.384(0.05) | 28.306(1.115) | 26.489(1.795) | **4.214(0.14)** | 4.402(0.253) | 27.363(1.036) | 4.258(0.206) |
| | Pearson ↑ | 0.866(0.002) | 0.934(0.012) | **0.951(0.004)** | 0.924(0.005) | 0.912(0.01) | 0.939(0.011) | 0.922(0.009) |
| | Spearman ↑ | 0.846(0.003) | 0.922(0.012) | **0.942(0.004)** | 0.911(0.006) | 0.897(0.013) | 0.931(0.011) | 0.907(0.01) |
| Parkinson (Motor) | MAE ↓ | 2.277(0.075) | 19.211(0.571) | 16.788(0.811) | 1.672(0.085) | 1.853(0.174) | 19.188(0.707) | **1.652(0.069)** |
| | RMSE ↓ | 3.927(0.036) | 19.687(0.583) | 19.373(0.752) | 2.701(0.131) | 3.116(0.361) | 19.709(0.938) | **2.68(0.134)** |
| | Pearson ↑ | 0.875(0.002) | 0.942(0.009) | **0.944(0.003)** | 0.942(0.005) | 0.919(0.022) | 0.943(0.009) | 0.943(0.006) |
| | Spearman ↑ | 0.868(0.002) | 0.939(0.009) | **0.94(0.004)** | 0.938(0.007) | 0.913(0.024) | **0.94(0.009)** | 0.939(0.006) |
| Super Conductivity | MAE ↓ | 6.577(0.051) | 20.445(0.88) | 14.27(0.631) | 6.293(0.072) | **6.238(0.051)** | 18.616(1.762) | 6.257(0.027) |
| | RMSE ↓ | 11.815(0.114) | 23.676(0.778) | 17.094(0.607) | **10.719(0.178)** | 10.935(0.042) | 21.514(1.661) | 10.745(0.183) |
| | Pearson ↑ | 0.94(0.001) | 0.95(0.002) | 0.947(0.001) | 0.95(0.002) | 0.948(0.0) | **0.951(0.001)** | 0.949(0.002) |
| | Spearman ↑ | 0.942(0.001) | 0.942(0.001) | 0.939(0.002) | 0.943(0.001) | **0.945(0.001)** | 0.942(0.001) | 0.944(0.002) |
| IC50 | MAE ↓ | **1.359(0.002)** | 3.815(0.009) | 3.549(0.044) | 1.369(0.006) | 1.362(0.012) | 3.736(0.021) | **1.359(0.003)** |
| | RMSE ↓ | 1.709(0.012) | 4.134(0.011) | 3.87(0.044) | 1.706(0.003) | 1.705(0.018) | 4.053(0.02) | **1.7(0.005)** |
| | Pearson ↑ | 0.041(0.045) | 0.146(0.022) | 0.293(0.034) | 0.07(0.062) | **0.313(0.023)** | **0.313(0.021)** | 0.302(0.028) |
| | Spearman ↑ | 0.085(0.035) | 0.152(0.034) | 0.267(0.019) | 0.115(0.031) | **0.268(0.009)** | 0.263(0.029) | 0.26(0.024) |
| Overall | Rank | 5.208 | 5.312 | 4.104 | 3.583 | 3.312 | 4.104 | 2.375 |

## B.9. Sensitivity Analysis

We include the sensitivity results for GAR on Concrete Compressive Strength, Wine Quality, Parkinson (Total), Parkinson (Motor), Super Conductivity as in Fig. 7,8,9,10,11, where we present the box-plot with mean and standard deviation from 5-fold-cross-validation. The evaluation procedure is identical as the main benchmark experiments stated in Appendix B.6 except that we fix different $\alpha$ for GAR. As we mentioned in the main content, the $\alpha$ is not very sensitive, but it is suggested to be tuned in [0.1,10] region.

## B.10. Different Batch Sizes

We conduct experiments on different batch sizes {32, 64, 128, 256, 512} for GAR. The experimental setting is kept as the same as the main experiments in Appendix B.6. The results are included in Tab. 6. Because the proposed GAR learns both the mean and variance of the label values by conventional loss and proposed pairwise losses (by Theorem 1), we additionally provide the convergences curves for the estimated means and standard deviations in Fig. 12. From both the table and the figures, we observe a strong correlation between more accurate estimation of mean and standard deviation to better testing performance (Batch Size = 32 on Concrete Compressive Strength and 64 on Parkinson Motor).

## B.11. Difference with Standardization

The formulation for $\mathcal{L}_{\text{diff}}^{\text{MSE}}$ and $\mathcal{L}_{\text{diffnorm}}^{p=2}$ in Eq. 5 and Eq. 7 might remind reader of data centering or standardization. In this subsection, we highlight the difference:

- $\mathcal{L}_{\text{diff}}^{\text{MSE}}$ and $\mathcal{L}_{\text{diffnorm}}^{p=2}$ 'centralize' or 'standardize' not only for ground truth label, but also for prediction. Besides, the predictions are dynamic during model training process.

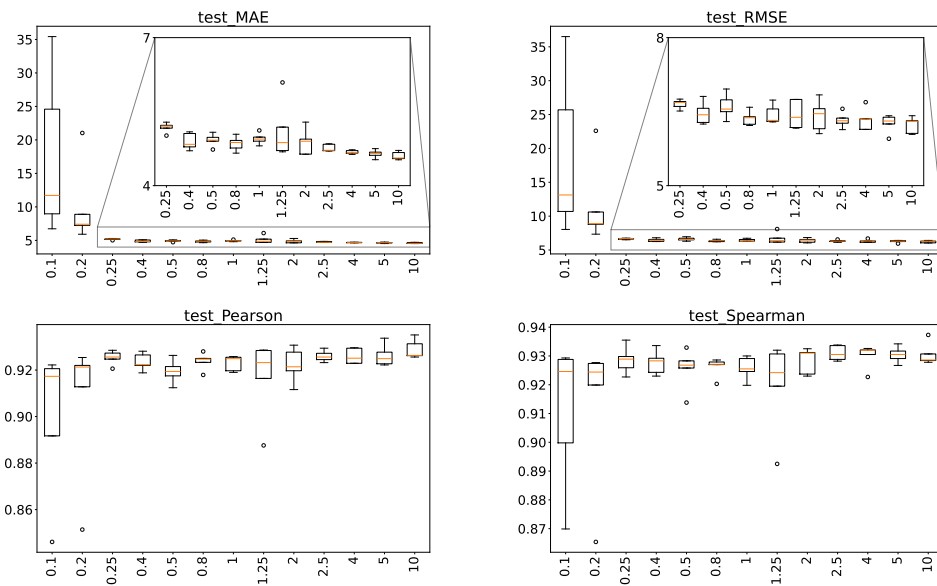

*Figure 7.* Different $\alpha$ (x-axis) for GAR on Concrete Compressive Strength.

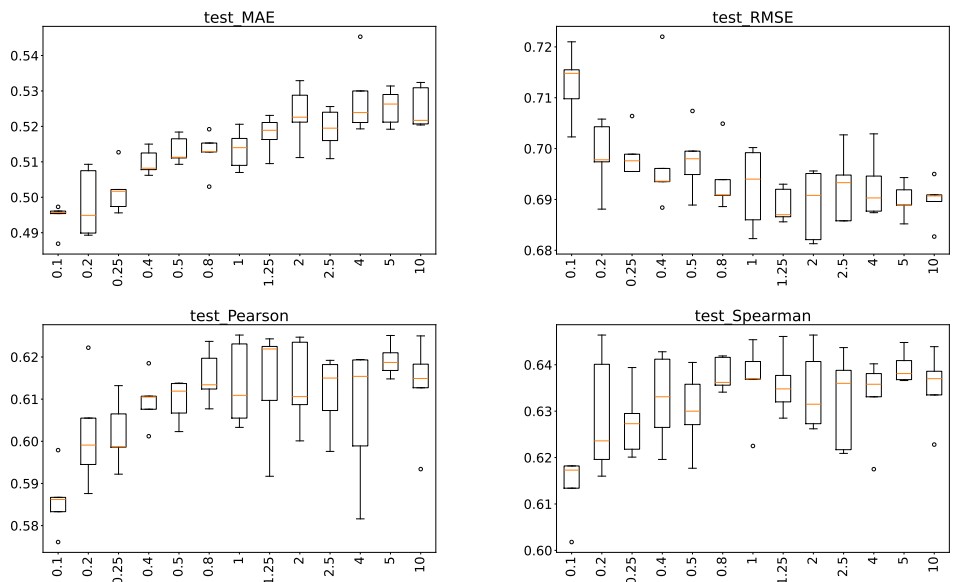

*Figure 8.* Different $\alpha$ (x-axis) for GAR on Wine Quality.

*Table 6.* Testing performance with different batch sizes.

| Dataset | Concrete Compressive Strength | | | | Parkinson(Total) | | | | Parkinson(Motor) | | | |
|---|---|---|---|---|---|---|---|---|---|---|---|---|
| Batch Size | MAE | RMSE | Pearson | Spearman | MAE | RMSE | Pearson | Spearman | MAE | RMSE | Pearson | Spearman |
| 32 | **3.866(0.132)** | **5.625(0.194)** | **0.942(0.005)** | **0.944(0.004)** | 1.875(0.306) | 3.957(1.684) | 0.925(0.063) | 0.944(0.014) | 1.518(0.074) | 2.661(0.201) | 0.944(0.008) | 0.942(0.007) |
| 64 | 3.966(0.183) | 5.705(0.257) | 0.939(0.005) | 0.937(0.009) | **1.431(0.109)** | **2.737(0.182)** | **0.967(0.004)** | **0.961(0.007)** | **1.184(0.033)** | **2.321(0.098)** | **0.958(0.004)** | **0.954(0.004)** |
| 128 | 4.167(0.113) | 5.837(0.178) | 0.937(0.004) | 0.938(0.002) | 1.61(0.075) | 2.971(0.12) | 0.961(0.003) | 0.954(0.003) | 1.315(0.06) | 2.455(0.097) | 0.953(0.004) | 0.949(0.005) |
| 256 | 4.603(0.075) | 6.222(0.148) | 0.929(0.004) | 0.931(0.003) | 2.64(0.071) | 4.258(0.206) | 0.922(0.009) | 0.907(0.01) | 1.652(0.069) | 2.68(0.134) | 0.943(0.006) | 0.939(0.006) |
| 512 | 4.989(0.139) | 6.568(0.226) | 0.917(0.006) | 0.921(0.008) | 4.032(0.056) | 5.679(0.04) | 0.855(0.007) | 0.83(0.007) | 2.709(0.062) | 3.849(0.126) | 0.891(0.016) | 0.883(0.018) |

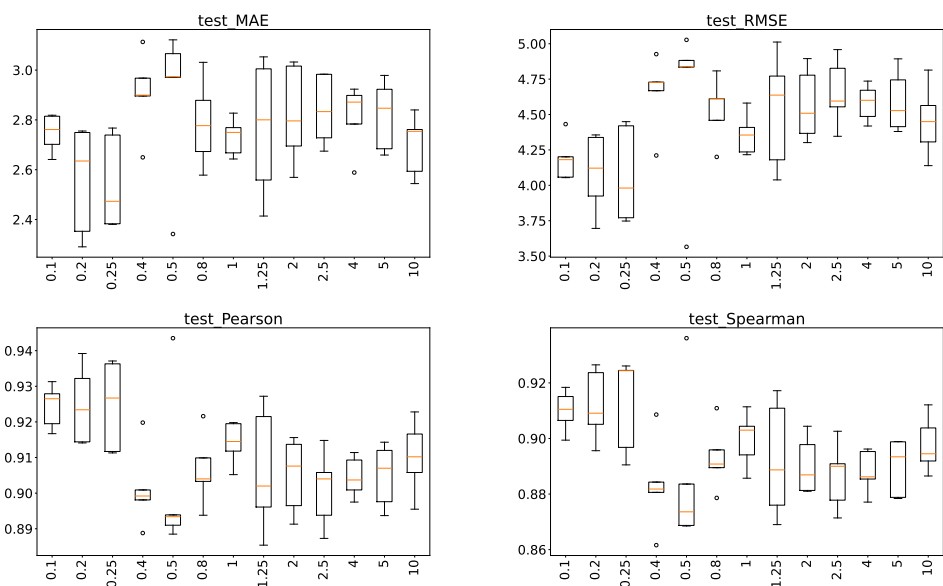

*Figure 9.* Different $\alpha$ (x-axis) for GAR on Parkinson (Total).

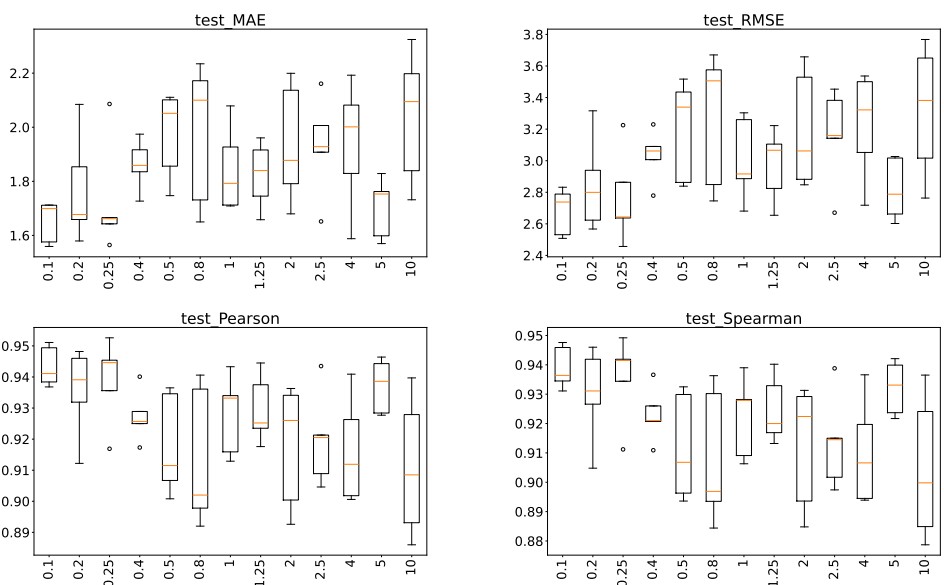

*Figure 10.* Different $\alpha$ (x-axis) for GAR on Parkinson (Motor).

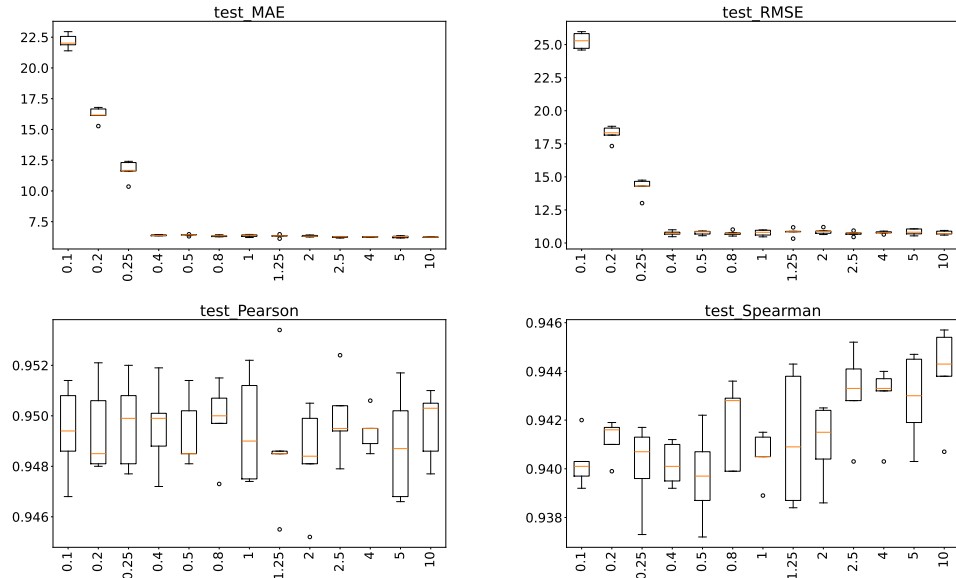

*Figure 11.* Different $\alpha$ (x-axis) for GAR on Super Conductivity.

- The proposed GAR, Alg. 1, learns the mean and standard deviation dynamically for each mini-batch during training, rather than for the whole dataset.

- We tested all the baselines on the five tabular datasets with standardization on both data features and labels, where the results are summarized in Tab. 7. From the results, the proposed GAR enjoys similar superiority on the standardized datasets as the previous main results in Tab. 1.

*Table 7.* Testing performance on the standardized tabular datasets/settings. Mean and standard deviation are reported. 'Gains' includes the percentages that GAR outperform the MAE and the best competitor (in parenthesis). The 'p-values' are calculated based on Student's t-test for the 'Gains'. p-value < 0.05 usually means significant difference. ↓ means the smaller the better; ↑ means the larger the better.

| Standardized Dataset | Method | MAE | MSE | Huber | Focal (MAE) | Focal (MSE) | RankSim | RNC | ConR | GAR (ours) | Gains (%) | p-values |
|---|---|---|---|---|---|---|---|---|---|---|---|---|
| Concrete Compressive Strength | MAE ↓ | 0.284(0.003) | 0.288(0.002) | 0.303(0.004) | 0.289(0.005) | 0.299(0.01) | 0.802(0.012) | 0.283(0.007) | 0.286(0.007) | **0.278(0.003)** | 2.22(1.77) | 0.02(0.24) |
| | RMSE ↓ | 0.39(0.02) | 0.387(0.009) | 0.396(0.01) | 0.386(0.008) | 0.391(0.014) | 0.994(0.005) | 0.38(0.002) | **0.378(0.006)** | 0.38(0.011) | 2.56(-0.66) | 0.41(0.7) |
| | Pearson ↑ | 0.924(0.003) | 0.919(0.005) | 0.918(0.002) | 0.921(0.004) | 0.921(0.004) | 0.674(0.059) | **0.925(0.003)** | **0.925(0.002)** | **0.925(0.004)** | 0.11(0.0) | 0.68(0.93) |
| | Spearman ↑ | 0.927(0.003) | 0.927(0.002) | 0.919(0.004) | 0.922(0.005) | 0.924(0.003) | 0.706(0.029) | **0.929(0.002)** | 0.927(0.005) | **0.929(0.002)** | 0.28(0.0) | 0.16(0.97) |
| Wine Quality | MAE ↓ | 0.574(0.003) | 0.611(0.006) | 0.615(0.003) | 0.59(0.001) | 0.62(0.005) | 0.595(0.005) | 0.616(0.005) | 0.579(0.005) | **0.565(0.007)** | 1.56(1.56) | 0.05(0.05) |
| | RMSE ↓ | 0.824(0.005) | 0.787(0.009) | 0.79(0.005) | 0.794(0.005) | **0.79(0.007)** | 0.817(0.009) | 0.81(0.003) | 0.811(0.006) | **0.79(0.004)** | 4.17(0.0) | 0.0(0.53) |
| | Pearson ↑ | 0.566(0.004) | 0.614(0.011) | 0.612(0.006) | 0.605(0.005) | 0.611(0.009) | 0.571(0.011) | 0.587(0.002) | 0.586(0.007) | **0.618(0.003)** | 9.22(0.67) | 0.0(0.5) |
| | Spearman ↑ | 0.603(0.007) | 0.63(0.009) | 0.63(0.006) | 0.624(0.003) | 0.623(0.006) | 0.603(0.006) | 0.609(0.005) | 0.616(0.005) | **0.638(0.007)** | 5.85(1.18) | 0.0(0.23) |
| Parkinson (Total) | MAE ↓ | 0.33(0.016) | 0.321(0.01) | 0.363(0.002) | 0.323(0.013) | 0.333(0.009) | 0.807(0.001) | 0.306(0.006) | 0.354(0.004) | **0.241(0.009)** | 26.81(21.12) | 0.0(0.0) |
| | RMSE ↓ | 0.507(0.013) | 0.473(0.012) | 0.515(0.003) | 0.505(0.011) | 0.487(0.014) | 10.828(19.644) | 0.474(0.015) | 0.524(0.006) | **0.385(0.02)** | 24.14(18.73) | 0.0(0.0) |
| | Pearson ↑ | 0.864(0.007) | 0.883(0.006) | 0.859(0.002) | 0.866(0.008) | 0.875(0.008) | 0.203(0.052) | 0.881(0.008) | 0.853(0.003) | **0.924(0.008)** | 6.91(4.63) | 0.0(0.0) |
| | Spearman ↑ | 0.841(0.007) | 0.86(0.006) | 0.835(0.003) | 0.845(0.006) | 0.852(0.006) | 0.188(0.043) | 0.861(0.009) | 0.826(0.009) | **0.909(0.008)** | 8.05(5.57) | 0.0(0.0) |
| Parkinson (Motor) | MAE ↓ | 0.309(0.017) | 0.313(0.017) | 0.362(0.004) | 0.314(0.01) | 0.339(0.02) | 0.847(0.002) | 0.313(0.01) | 0.339(0.011) | **0.225(0.014)** | 27.13(27.13) | 0.0(0.0) |
| | RMSE ↓ | 0.487(0.028) | 0.46(0.032) | 0.514(0.004) | 0.495(0.01) | 0.486(0.026) | 0.993(0.003) | 0.477(0.014) | 0.512(0.011) | **0.37(0.02)** | 24.05(19.58) | 0.0(0.0) |
| | Pearson ↑ | 0.871(0.015) | 0.886(0.016) | 0.856(0.002) | 0.868(0.005) | 0.872(0.014) | 0.13(0.022) | 0.878(0.007) | 0.857(0.007) | **0.928(0.008)** | 6.51(4.72) | 0.0(0.0) |
| | Spearman ↑ | 0.864(0.016) | 0.872(0.018) | 0.847(0.001) | 0.861(0.006) | 0.865(0.015) | 0.133(0.046) | 0.866(0.007) | 0.847(0.006) | **0.923(0.009)** | 6.79(5.81) | 0.0(0.0) |
| Super Conductivity | MAE ↓ | 0.244(0.001) | 0.256(0.0) | 0.263(0.001) | 0.247(0.001) | 0.264(0.002) | 0.574(0.068) | 0.241(0.001) | 0.243(0.001) | **0.223(0.002)** | 8.85(7.59) | 0.0(0.0) |
| | RMSE ↓ | 0.396(0.002) | 0.389(0.001) | 0.399(0.001) | 0.394(0.003) | 0.392(0.002) | 0.762(0.083) | 0.385(0.001) | 0.394(0.001) | **0.358(0.003)** | 9.68(7.02) | 0.0(0.0) |
| | Pearson ↑ | 0.919(0.0) | 0.921(0.001) | 0.917(0.0) | 0.92(0.001) | 0.92(0.001) | 0.811(0.008) | 0.923(0.001) | 0.92(0.0) | **0.934(0.001)** | 1.66(1.14) | 0.0(0.0) |
| | Spearman ↑ | 0.916(0.001) | 0.907(0.001) | 0.901(0.001) | 0.912(0.001) | 0.9(0.002) | 0.825(0.004) | 0.916(0.001) | 0.917(0.001) | **0.928(0.001)** | 1.25(1.17) | 0.0(0.0) |
| IC50 | MAE ↓ | 0.82(0.002) | 0.829(0.003) | 0.821(0.002) | 0.821(0.002) | 0.831(0.002) | 0.82(0.002) | 0.827(0.002) | 0.814(0.001) | **0.79(0.017)** | 3.7(2.97) | 0.01(0.02) |
| | RMSE ↓ | 1.026(0.002) | 1.016(0.001) | 1.026(0.002) | 1.025(0.002) | 1.021(0.002) | 1.031(0.005) | 1.026(0.002) | 1.024(0.003) | **0.992(0.024)** | 3.33(2.35) | 0.02(0.08) |
| | Pearson ↑ | 0.079(0.025) | 0.166(0.027) | 0.122(0.031) | 0.089(0.019) | 0.156(0.021) | 0.015(0.025) | -0.012(0.017) | 0.224(0.031) | **0.3(0.035)** | 278.81(33.82) | 0.0(0.01) |
| | Spearman ↑ | 0.13(0.031) | 0.165(0.032) | 0.12(0.028) | 0.11(0.02) | 0.146(0.021) | 0.026(0.041) | 0.019(0.042) | 0.198(0.024) | **0.277(0.027)** | 112.94(39.9) | 0.0(0.0) |

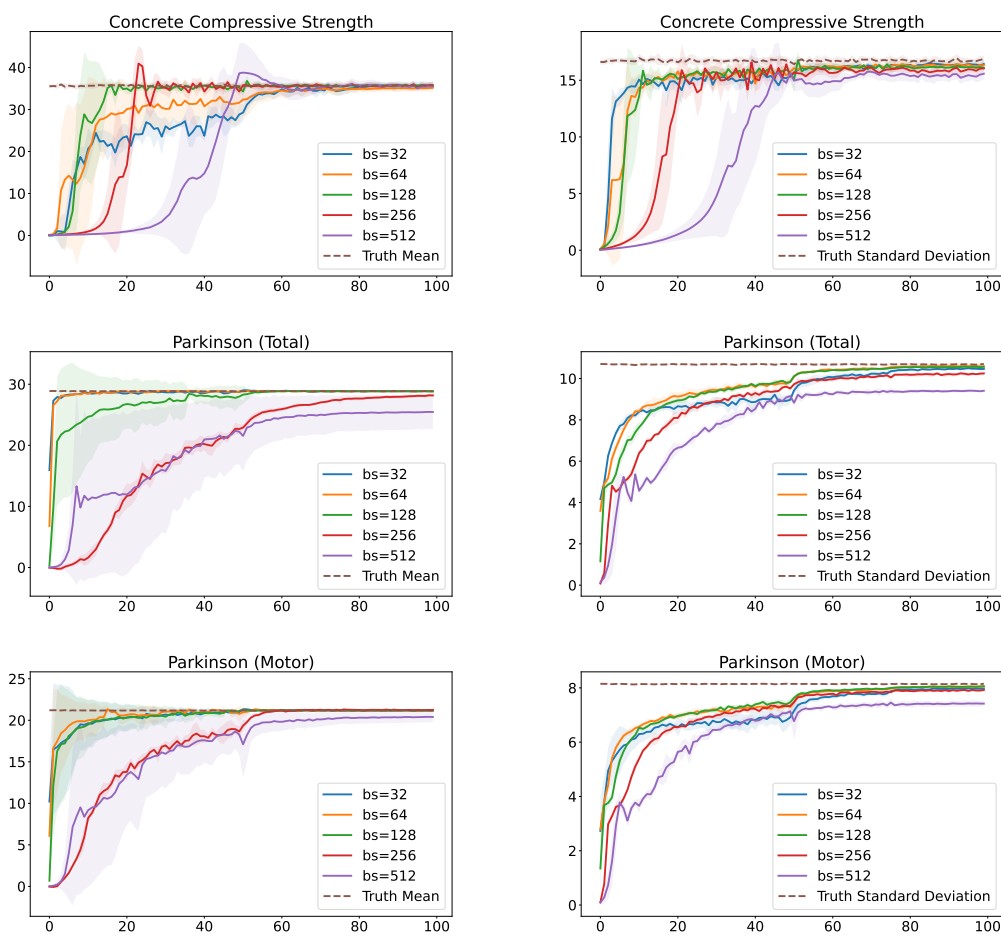

*Figure 12.* Batch Sizes on mean and standard deviation estimation for GAR

