# OpenReview forum: "Gradient Aligned Regression via Pairwise Losses"
_ICML.cc/2025/Conference — ICML 2025 poster_

### Official Review · Reviewer_DxkR · 2025-02-22

**Overall Recommendation:** 3

**Summary:**

This paper proposes GAR for solving regression problems while maintaining label similarity. The authors introduce two pairwise label difference losses and explain their theoretical insights. Various experiments and ablation studies demonstrate that GAR outperforms other baselines.

**Claims And Evidence:**

The claims are supported by authors' theoretical analysis and experiments.

**Essential References Not Discussed:**

Related works are well discussed as far as I am concerned.

**Experimental Designs Or Analyses:**

The experiments mainly focus on test set loss and statistical correlations. Extensive ablation experiments validate the effectiveness of GAR. The only drawback is that the combination of the three loss functions relies on a heuristic method and is sensitive to hyperparameters.

**Methods And Evaluation Criteria:**

The authors introduce two pairwise label difference losses and explain their theoretical insights. They also propose a DRO-based method to combine the two losses into a single objective function. The experiments mainly focus on test set loss and statistical correlations, and the results show that GAR can improve the model's fitting ability.

**Other Comments Or Suggestions:**

1. $\mathcal{L}\_{\text {diff }}^{M S E}$ in Theorem 1 is not defined.
2. $L_{\mathrm{diff}}$ should be $\mathcal{L}\_{\text {diff }}$  at Line 111.

**Other Strengths And Weaknesses:**

Strengths
1. This paper is well-written and easy to follow.
2. Numerical experiments and ablation studies are extensive and demonstrate the advantages of the method in fitting the data distribution.

Weaknesses
1. The proposed method mainly focuses on the clean data setting, while other baselines in the experiments, for example, RNC, have applications in improving generalization and robustness.
2. $\mathcal{L}\_{\text{diffnorm}}$ is claimed to be a relaxed version of $\mathcal{L}\_{\text{diff}}$. It is unclear why the objective function includes both $\mathcal{L}\_{\text{diffnorm}}$ and $\mathcal{L}\_{\text{diff}}$, rather than just  $\mathcal{L}\_{\text{diffnorm}}$.

**Questions For Authors:**

See strengths and weaknesses.

**Relation To Broader Scientific Literature:**

As the authors point out in the Related Work section, their proposed method has applications in contrastive learning.

**Theoretical Claims:**

The authors mainly prove that the two proposed losses are equivalent to the variance of prediction errors and the negative Pearson correlation coefficient between predictions and ground truth, as well as the theoretical insight between learning pairwise label difference and learning the gradients of ground truth function. Proofs seem correct to me.

---

> ### Author Rebuttal · Authors · 2025-03-30
>
> Thank you for your invaluable comments! Here are our responses:
>
> 1. "*The proposed method mainly focuses on the clean data setting, while other baselines in the experiments, for example, RNC, have applications in improving generalization and robustness.*"
>
> 	**Response**: RNC is a nice regression method that incorporates pairwise label similarities with contrastive learning style method in latent feature space. It enjoys better generalization and robustness with $\delta$-ordered feature embedding, which regularizes hypothesis space to be smaller. Although we limit our scope on clean data setting, we want to highlight the following points:
>
> 	i) The proposed GAR is a novel loss function for regression and RNC is a kind of regularization on latent feature space. Those methods regularizing the latent feature space, such as RNC, are complementary to GAR, as our results on AgeDB (Table 1) demonstrate that GAR with RNC can enjoy better performance.
>
> 	ii) Different with the latent feature space based methods, GAR is proposed with more focus on the computational efficiency and theoretical insights (Theorem 1, Corollary 3 and Theorem 4), which demonstrates the unique advantages for GAR among the other pairwise regression methods.
>
> 2.  "*$\mathcal L_{\text{diffnorm}}$ is claimed to be a relaxed version of $\mathcal L_{\text{diff}}$. It is unclear why the objective function includes both $\mathcal L_{\text{diffnorm}}$ and $\mathcal L_{\text{diff}}$, rather than just $\mathcal L_{\text{diffnorm}}$*"
>
> 	**Response**: As the Theorem 4 shows that fitting the label difference equals to fitting the gradients of the underlying ground truth function, therefore $\mathcal L_{\text{diff}}$ is optimized to fit $\nabla\mathcal Y$ and the relaxed version $\mathcal L_{\text{diffnorm}}$ is optimized to fit the normalized $\nabla\mathcal Y$ (captures the function shape).
>
> 	If we only include $\mathcal L_{\text{diffnorm}}$ to fit the normalized $\nabla\mathcal Y$, the objective function only could capture the function shape of $\mathcal Y$ and could lose the mechanism on capturing the original $\nabla\mathcal Y$ with magnitude information on gradients.
>
> 	On the other hand, adding $\mathcal L_{\text{diffnorm}}$ doesn't involve any extra hyper-parameter tuning burden, as we balance all the terms by a DRO framework (Theorem 5). Besides, our ablation study (Table 5 in the appendix) shows that including both $\mathcal L_{\text{diff}}$ and $\mathcal L_{\text{diffnorm}}$ achieves the best overall performance. Only including $\mathcal L_{\text{diffnorm}}$ is the suboptimal (although it is the second best).
>
> 3. "*Other Comments Or Suggestions: 1. $\mathcal L_{\text{diff}}^{\text{MSE}}$ in Theorem 1 is not defined. 2. $L_{\text{diff}}$ should be $\mathcal L_{\text{diff}}$ at Line 111.*"
>
> 	**Response**: We appreciate your careful checks on our writing to improve the readability.
>
> 	We will update $\mathcal L_{\text{diff}}^{\text{MSE}}\coloneqq\frac{1}{N^2}\sum_{i=1}^{N}\sum_{j=1}^N \ell_{\frac{1}{2}\text{MSE}}(f(\mathbf x_i)-f(\mathbf x_j), \mathcal Y(\mathbf x_i)-\mathcal Y(\mathbf x_j))$ right above Eq 4.    We will correct the typo at Line 111 as $\mathcal L_{\text{diff}}$.
>
>
> Thank you again for providing thoughtful comments! Please let us know if our responses address your concerns.

---

### Official Review · Reviewer_pc7o · 2025-03-12

**Overall Recommendation:** 3

**Summary:**

This paper proposes to use pairwise comparison to improve regression accuracy. Authors propose to focus on the difference of pairs, and propose two loss terms comparing the differences of pairs. Authors then show the equivalence form that enjoys linear computational efficiency using specific loss functions. Furthermore, authors propose a reformulation to reduce the number of hyperparameters to only one, and a computational stability trick. Extensive empirical evaluations are conducted to show the efficiency and improvement of the proposed method.

**Claims And Evidence:**

The three claims in the paper, as well as two advantage points listed in introduction are well supported by the demonstration.

**Essential References Not Discussed:**

No to the best of my knowledge.

**Experimental Designs Or Analyses:**

The classical regression experiment setting is conducted and the implementation codes in python is also provided. Ablations studies including the only hyperparameter are properly conducted.

**Methods And Evaluation Criteria:**

The proposed methods improve regression accuracy, and common regression measures are used for evaluation.

**Other Comments Or Suggestions:**

N/A

**Other Strengths And Weaknesses:**

For strengths, the paper is well self-contained by proposing a pragmatic method with theoretical support and extensive empirical evaluations. It also clearly demonstrates its limit of considering single scalar regression with no noise.
However, the significance of how the proposed method is located in the literature is not very clear for readers who's not familiar with the specific literature of using pairwise data for regression. For example, being said "However, these methods heuristically approximate label similarity information as a relative rank given an anchor data point, which may lead to the loss of some original pairwise quantitative information.", it would be desirable to have more intuitive demonstration on the weakness of existing methods. What does the anchor data point mean in this case?

**Questions For Authors:**

I have listed my questions in above comments.

**Relation To Broader Scientific Literature:**

As mentioned in the paper, this paper related to the broad regression literature, and shows the efficient utilization of pairwise data on improving regression accuracy.

**Theoretical Claims:**

I have checked the presentation of theoretical claims and their brief prood sketch in the main paper.

---

> ### Author Rebuttal · Authors · 2025-03-30
>
> We sincerely thank you for your invaluable feedback on our manuscript! Here are our responses:
>
> 1. "*However, the significance of how the proposed method is located in the literature is not very clear for readers who's not familiar with the specific literature of using pairwise data for regression. For example, being said "However, these methods heuristically approximate label similarity information as a relative rank given an anchor data point, which may lead to the loss of some original pairwise quantitative information.", it would be desirable to have more intuitive demonstration on the weakness of existing methods. What does the anchor data point mean in this case?*"
>
> 	**Response**: Thank you for pointing out this for us to improve the readability for the broader audience. We plan to modify this part on introduction as follows:
>
> 	For those latent space based pairwise regression methods, they define an 'anchor data point', which either act as a reference point for calculating the relative ranks to the anchor for the other data (Gong et al., 2022), or the ranks are consequently utilized to construct positive and negative pairs (Zha et al.,2023; Keramati et al., 2023). We refer the readers to the related work section (line 62, right column) for more information about those methods.
>
> 	One of the imperfectnesses for those approaches is conceptually to see: the original pairwise label similarities (like the pairwise label differences in our work) are converted to ranks or positive/negative pairs. There is an approximation loss because the conversion is single directional (and the original label similarities cannot be recovered from the coarser converted information).
>
> Please let us know if our explanation addresses your question. Thank you again for your useful comments!

---

> > ### Comment · Reviewer_pc7o · 2025-04-08
> >
> > I thank authors for their detailed response and choose to lean on accepting the paper.

---

> > > ### Author Response · Authors · 2025-04-09
> > >
> > > Thanks for your follow-up and for taking the time to consider our responses. We're glad they were helpful and appreciate your support.

---

### Official Review · Reviewer_qJh5 · 2025-03-24

**Overall Recommendation:** 3

**Summary:**

In this work, the authors focus on the problem of regression with pairwise losses. Firstly, it is revealed that the conventional regression losses (e.g., MSE and MAE) ignore the pairwise information of training samples and can have larger error variance. Inspired by the success of existing pairwise losses, the authors provide a loss that penalizes the mismatch between prediction and target's pairwise difference instead of pointwise deviation, which can be computed in linear time w.r.t. sample size. A relaxed version is also provided, which allows the mismatch of pairwise difference's magnitude. An aggregation scheme is also provided to combine the conventional loss, pairwise loss, and the relaxed version with only one hyperparameter. Finally, the intuition behind learning pairwise difference is shown by its equivalence to learning the ground truth function's gradient. Experimental results demonstrate the effectiveness of the proposed methods on various regression benchmark datasets.

**Claims And Evidence:**

The claims in this submission are formulated as theorems and are supported by rigorous proof.

**Essential References Not Discussed:**

The cited works primarily focus on regression. However, similar types of loss functions have also been explored in the context of AUC optimization. For example, [1] investigates the consistency of AUC pairwise optimization, which is relevant to the discussion and it is encouraged to consider them for a more comprehensive review of related methodologies.

[1]. Gao W, Zhou Z H. On the Consistency of AUC Pairwise Optimization. IJCAI, 2015.

**Experimental Designs Or Analyses:**

A detailed ablation study is conducted, including sensitivity analyses on various hyperparameters. Additionally, the running time analysis experimentally confirms the linear time complexity of the proposed method.

**Methods And Evaluation Criteria:**

The proposed method is compared with state-of-the-art on various real-world datasets, which make sense for the problem of regression.

**Other Comments Or Suggestions:**

Please see the questions.

**Other Strengths And Weaknesses:**

Strengths:

1. The result of Theorem 4 is non-trivial, which reveals the rationale of learning pairwise difference.

2. The writing is clear and easy to follow, and the motivation of each proposed method are elaborated.

Weakness:

Please see the questions.

**Questions For Authors:**

In this paper, the target is assumed to be a deterministic function $\mathcal{Y}$. This assumption is more restrictive than the commonly considered stochastic function setting. Generalizing the results to the stochastic case appears to be non-trivial, as key conclusions (e.g., Theorem 4) rely on this deterministic assumption.

**Relation To Broader Scientific Literature:**

The relationship between this work and previous studies on pairwise losses for regression is discussed, with a particular focus on contrastive learning-based methods.

**Theoretical Claims:**

I have examined the proofs of Theorem 1, Corollary 2, and Corollary 3 and found no apparent flaws.

---

> ### Author Rebuttal · Authors · 2025-03-30
>
> We appreciate your invaluable comments for our manuscript! Here are our responses:
>
> 1. "*The cited works primarily focus on regression. However, similar types of loss functions have also been explored in the context of AUC optimization...*"
>
> 	**Response**: we sincerely thank you for raising this point and we agree that the formulations share a similarity between the proposed pairwise regression losses and the pairwise losses utilized for AUROC optimization. Nevertheless, we will also highlight the difference: the propose pairwise regression losses do not define positive/negative pairs and the target is continuous. We will add the discussion in the related work section with the recommended reference.
>
> 2. "*In this paper, the target is assumed to be a deterministic function $\mathcal Y$. This assumption is more restrictive than the commonly considered stochastic function setting. Generalizing the results to the stochastic case appears to be non-trivial, as key conclusions (e.g., Theorem 4) rely on this deterministic assumption.*"
>
> 	**Response**: Thank you for pointing out the implicit assumption and the potential extension on our current Theorem 4. Yes, its current proof relies on the assumption that $\mathcal Y(\mathbf x)$ is deterministic. We will explicitly add the assumption in our updated manuscript. It is non-trivial to extend to arbitrarily general stochastic $\mathcal Y(\mathbf x)$ setting but we're happy to provide the following extensive discussion and an extended theorem that works on the stochastic function under certain forms (should cover the "*commonly considered stochastic function setting*" as you mentioned) by expectation trick.
>
> 	If $\mathcal Y(\mathbf x)$ is stochastic, the $\nabla\mathcal Y(\mathbf x)$ would also be stochastic. Therefore, the deterministic model $\nabla f(\mathbf x)$ inherently could not capture stochastic $\nabla\mathcal Y(\mathbf x)$. A simple example could be $\mathcal Y(\mathbf x, z) = m(\mathbf x) +z\cdot s(\mathbf x)$, where $z\sim N(0, 1)$ is stochastically sampled from the standard normal distribution; $m(\mathbf x)$ and $s(\mathbf x)$ represent the ground truth mean and standard deviation for general heteroskedastic case. In this case, the fitting model have to learn with the randomness (like the reparameterization trick in VAE) in order to capture the stochastic $\nabla\mathcal Y(\mathbf x, z)$. Similarly as Theorem 4, we can try to define the extended theorem:
> 	$f(\mathbf x_1, z_1) - f(\mathbf x_2, z_2)=\mathcal Y(\mathbf x_1, z_1) - \mathcal Y(\mathbf x_2, z_2), \forall (\mathbf x_1, \mathcal Y(\mathbf x_1, z_1)), (\mathbf x_2, \mathcal Y(\mathbf x_2, z_2))\in\mathcal D; \forall z_1,z_2\sim N(0,1),$
> 	**iif**
> 	$\nabla^kf(\mathbf x, z)=\nabla^k\mathcal Y(\mathbf x, z),\forall (\mathbf x, \mathcal Y(\mathbf x, z))\in\mathcal D; \forall z\sim N(0,1)$.
>
> 	We still could use the same logic for the proof by treating $z$ as an extra variable. However, the difficulty is learning the stochastic $z\cdot s(\mathbf x)$ term for the model because $z$ is unknown during training (or even assume the distribution for $z$ is known, the sampled value is still unknown). $f(\mathbf x, z)$ may not converge to $\mathcal Y(\mathbf x, z)$.
>
> 	**However, we could be more conservative and only argue the model can capture the expected gradient on $\mathbf x$**, i.e. $\mathbb E_z\big[\nabla_{\mathbf x}\mathcal Y(\mathbf x, z)\big]$, where deterministic model $f(\mathbf x)$ is competent. We can try to define the extended theorem:
> 	$f(\mathbf x_1) - f(\mathbf x_2)=\mathbb E_{z_1}\big[\mathcal Y(\mathbf x_1, z_1)\big]- \mathbb E_{z_2}\big[\mathcal Y(\mathbf x_2, z_2)\big], \forall (\mathbf x_1, \mathcal Y(\mathbf x_1, z_1)), (\mathbf x_2, \mathcal Y(\mathbf x_2, z_2))\in\mathcal D; \forall z_1,z_2\sim N(0,1),$
> 	**iif**
> 	$\nabla^kf(\mathbf x)=\mathbb E_z\big[\nabla^k_{\mathbf x}\mathcal Y(\mathbf x, z)\big],\forall (\mathbf x, \mathcal Y(\mathbf x, z))\in\mathcal D; \forall z\sim N(0,1)$.
>
> 	Notice that $\mathbb E_{z}\big[\mathcal Y(\mathbf x, z)\big]=m(\mathbf x)$, $\mathbb E_z\big[\nabla^k_{\mathbf x}\mathcal Y(\mathbf x, z)\big]=\nabla^km(\mathbf x)$. Our previous proof still goes through by replace $\mathcal Y(\mathbf x)$ as $m(\mathbf x)$. Actually, this extended theorem works for any stochastic function as far as the stochastic term can be decoupled as 0 mean and can be canceled under expectation.
>
> 	We will add this discussion and extended theorem to the updated manuscript!
>
>
> Thanks again for your constructive comments! Please let us know if our responses address your questions.

---

### Decision · Program_Chairs · 2025-05-01

**Decision:**

Accept (poster)

**Comment:**

This paper studies regression by incorporating label similarity information to mitigate the estimation variance. All reviewers agree on that the paper is well presented and the authors' claims are verified thoroughly from theory and practice. Thus I accept this submission for acceptance.